

# Diurnal fluxes of HONO above a crop rotation

**Sebastian Laufs[1], Mathieu Cazaunau[2,3], Patrick Stella[4,5], Ralf Kurtenbach[1], Pierre Cellier[4], Abdelwahid Mellouki[2], Benjamin Loubet[4] and Jörg Kleffmann[1]**

[1] Physikalische und Theoretische Chemie, Fakultät 4, Bergische Universität Wuppertal, Gaußstraße 20, 42119 Wuppertal, Germany

[2] ICARE-CNRS, 1 C Av. de la Recherche Scientifique, 45071 Orléans cedex 2, France

[3] LISA, UMR 7583, CNRS, Universités Paris Est Créteil et Paris Diderot, 94010 Créteil, France

[4] UMR ECOSYS, INRA, AgroParisTech, Université Paris-Saclay, 78850, Thiverval-Grignon, France

[5] UMR SADAPT, AgroParisTech, INRA, Université Paris-Saclay, 75005, Paris, France

*Correspondence to*: Jörg Kleffmann (*kleffman@uni-wuppertal.de*)

**Abstract.** Nitrous acid (HONO) fluxes were measured above an agricultural field site near Paris during different seasons, above bare soil and different crops using the aerodynamic gradient (AG) method. Two LOPAPs (LOng Path Absorption Photometer) were used to determine the HONO gradients between two heights. During daytime mainly positive HONO fluxes were observed which showed strong correlation with the product of the $NO_2$ concentration and the long wavelength UV light intensity, expressed by the photolysis frequency $J(NO_2)$. These results indicate HONO formation by photosensitized heterogeneous conversion of $NO_2$ on soil surfaces as observed in recent laboratory studies. An additional influence of the soil temperature on the HONO flux can be explained by the temperature dependent HONO adsorption on the soil surface. A parameterization of the HONO flux at this location with $NO_2$ concentration, $J(NO_2)$, soil temperature and humidity fits reasonably well all flux observations at this location.

## 1   Introduction

During the last decades, many field measurement campaigns have reported unusually high nitrous acid (HONO) concentrations during daytime, for remote (Zhou et al., 2002; Acker et al., 2006a; Sörgel et al., 2011a; Villena et al., 2011; Oswald et al., 2015; Meusel et al., 2016), semi-urban (Neftel et al., 1996; Staffelbach et al., 1997; Kleffmann et al., 2005; Yang et al., 2014) and urban regions (Kleffmann et al., 2002; 2003; Ren et al., 2003; Acker et al., 2006b; Ren et al., 2006; Elshorbany et al., 2009; 2010; Hou et al., 2016; Lee et al., 2016). These results stimulated laboratory investigations on potential HONO precursors from which the most frequently discussed mechanisms are (i) the photosensitized reduction of nitrogen dioxide ($NO_2$) by organic material, e.g. humic acids (George et al., 2005; Stemmler et al., 2006; 2007; Sosedova et al., 2011; Han et al. 2016), (ii) the photolysis of adsorbed nitric acid (Zhou et al., 2003; 2011; Laufs and Kleffmann, 2016), (iii) bacterial production of nitrite in soil (Su et al., 2011; Ostwald et al., 2013; Maljanen et al., 2013; Oswald et al., 2015; Scharko et al., 2015; Weber, 2015) and (iv) release of adsorbed HONO from soil surfaces after deposition of strong acids (VandenBoer et al., 2013; 2014; 2015; Donaldson et al., 2014). Another discussed source, the reaction of excited gaseous $NO_2$ with water (Li et al., 2008), is of minor importance as demonstrated by laboratory (Crowley and Carl, 1997; Carr et al., 2009; Amedro et al., 2011) and modelling studies (Sörgel et al.,





2011b; Czader et al., 2012). Also the photolysis of nitro-phenols or similar compounds (Bejan et al., 2006) is meaningful only in polluted areas, where concentrations of these precursors are high. Finally, the gas-phase HONO source by the reaction of $HO_2 \times H_2O$ complexes with $NO_2$, recently postulated by Li et al. (2014), could not be confirmed by the same group in simulation chamber experiments (Li et al., 2015) and is also in conflict

with recent aircraft measurements (Ye et al., 2015).

Several field studies point to an atmospheric daytime HONO source by heterogeneous photosensitized reduction of $NO_2$ on organic substrates (Kleffmann, 2007). In these studies, calculated daytime HONO sources, determined from HONO levels exceeding theoretical photostationary state (PSS) values, showed high correlations with the photolysis rate coefficient $J(NO_2)$ or the irradiance and $NO_2$ concentration (Elshorbany et

al., 2009; Sörgel et al. 2011b; Villena et al., 2011; Wong et al., 2012; Lee et al., 2016). However, concentrations are not only controlled by the local ground surfaces source processes, but depend also on the convective mixing in the atmosphere leading to a potential misinterpretation of the correlation results. In addition, the assumed PSS conditions may also not be fulfilled when HONO and its precursors were measured close to their sources (Lee et al., 2013).

In contrast, flux measurements are able to give direct information about ground surface production and loss processes and are potentially a better tool to investigate HONO sources in the lower atmosphere. Nowadays, eddy covariance (EC) is the most commonly applied method to measure fluxes between the surface and the atmosphere. The lack of fast and sensitive EC-HONO measurement systems, however, requires the use of indirect methods like the aerodynamic gradient (AG) method that has been described by a number of authors

(Thom et al., 1975; Sutton et al., 1993) or the relaxed eddy accumulation method (REA) that was recently used also for HONO (Ren et al., 2011; Zhou et al., 2011; Zhang et al., 2012). Unfortunately, the available flux observations indicate different HONO precursors. Harrison and Kitto (1994) and Ren et al. (2011), for example, found a relationship of the HONO flux with the $NO_2$ concentration and also its product with light intensity, whereas Zhou et al. (2011) observed a correlation of the HONO flux with adsorbed nitric acid and short

wavelength radiation. A campaign above a grassland spread with manure (Twigg et al., 2011) found no evidence for a $NO_2$ driven mechanism producing upward HONO fluxes at a local field site, although HONO and $NO_2$ concentrations were coupled with one another, which indicated a regional connection. Hence, the origin of the ground surface daytime HONO source is still a topic of controversial discussion.

The present study was part of the German-French (DFG / INSU-CNRS) PHOTONA project (PHOTOlytic

sources of Nitrous Acid in the atmosphere) with laboratory and field investigations concerning HONO in the troposphere. In the present work only the field campaigns aimed at elucidating unknown sources of HONO by flux measurements above an agricultural field site are described. The measurements were performed during different seasons of the year and above different types of canopies using the aerodynamic gradient (AG) method and the LOPAP (LOng Path Absorption Photometer) technique.

## 2    Methods

### 2.1.    Field Site

The measurement site is an agricultural field located in Grignon, around 40 km west from Paris, France (48.9 N, 1.95 E). This site, operated by INRA (Institut National de la Recherche Agronomique) and hosted by





AgroParisTech, was part of the EC CarboEurope-IP, NitroEurope-IP, Eclaire-IP and INGOS-IP European
projects and is part of the ICOS and Fluxnet measurement networks. The site is well documented (Laville et al., 2009; Loubet et al., 2011) as are several experiments on reactive trace gases performed at the site (Bedos et al., 2010; Loubet et al., 2012; 2013; Potier et al., 2015; Wu et al., 2015; Personne et al., 2015; Vuolo et al., 2016). Briefly, measurements were carried out on a 19 ha field with a fetch of 100-400 m depending on wind direction. Roads with substantial traffic surround the site to the south (700 m), east (900 m), north-west (200 m) and south-west (500 m). Other agricultural fields surround the site to the north, south and east. The small village Grignon is located to the west around 700 m away from the measurement site. An animal farm with an average annual production of 210 cattle, 510 sheep and 900 lambs is situated 400 m to the south-south west. The soil on the field is a silt loam with 31 % clay, 62.5 % silt and 6.5 % sand and was managed with a maize, winter wheat, winter barley, mustard rotation. The field is annually fertilised with nitrogen solution and cattle manure at a rate varying between 100 and 300 kg N $ha^{-1}$ $y^{-1}$, with manure usually applied every 2 to 3 years.

### 2.2. Experimental design

Three field campaigns were performed during the PHOTONA project over a range of crop developments and types: PHOTONA 1 was carried out from 20th to 30th August 2009 over bare soil (for more details see Stella et al., 2012). PHOTONA 2 was a spring campaign from 7th to 19th April 2010 following fertilisation with nitrogen solution (composed of 50 % of ammonium nitrate and 50 % of urea, with a nitrogen content of 39 % in dry mass) on 17th March (60 kg N $ha^{-1}$) and 6th April (40 kg N $ha^{-1}$) over a growing (from 0.2 to 0.3 ± 0.05 m height) triticale (hybrid of wheat -*Triticum*- and rye -*Secale*-) canopy seeded on 14th October 2009. PHOTONA 3 was a second summer campaign from 16th to 30th August 2011 over a well-developed maize canopy of about 2 m height, seeded on 21st April 2011. The last fertilisation of the site before the campaign was on 17th March 2011 with cattle slurry application of 25 $m^3$ $ha^{-1}$. The slurry composition was 10.5 % dry matter, 28.9 g N $kg^{-1}$ of dry matter (of which 12.4 g N $kg^{-1}$ was ammonium), corresponding to an amount of nitrogen of 33.4 kg N $ha^{-1}$. During all campaigns HONO mixing ratios were measured at two heights above the canopy using the LOPAP technique (QUMA Elektronik & Analytik GmbH, Germany) which is explained in detail elsewhere (Heland et al., 2001; Kleffmann et al., 2002). The LOPAP instrument allows detection of HONO down to mixing ratios of 1 pptV and the instrument showed excellent agreement with the DOAS (Optical Absorption Spectroscopy) technique during intercomparison studies (Kleffmann et al., 2006). Recently, 15 % interference against $HNO_4$ was inferred from laboratory experiments for the LOPAP instrument (Legrand et al., 2014). However, because of the typical high temperatures and low $NO_2$ levels during daytime, low $HNO_4$ levels (<50 ppt) are expected for the present study, leading to no significant overestimation of the HONO data. Two LOPAP instruments were placed in thermostated field racks, with the external sampling units fixed at two heights on a mast in the open atmosphere (see Figure 1). Other trace gases measured during the campaigns were $NO_2$ that was detected with the sensitive Luminol technique (LMA-3D or LMA-4, Unisearch Associates Inc., Ontario, Canada), NO (CLD780TR, Ecophysics, Switzerland) and $O_3$ (FOS, Sextant Technology Ltd, New Zealand), which are based on chemiluminescence techniques. Since the Ecophysics $NO_x$ instrument is relatively slow and measures NO and $NO_x$ consecutively, only the NO channel was used while $NO_2$ was detected by the Luminol instrument. Here potential overestimation of $NO_2$ by interferences against peroxyacyl nitrates was ignored since their concentrations were unknown. $NO_2$, NO and $O_3$ were sampled at a flow rate exceeding 40 L $min^{-1}$ with 7 m





(9.24 mm internal diameter (i.d.)) plus 7 m (3.96 mm i.d.) long PFA (Perfluoroalkoxy polymer) sampling lines during PHOTONA 1, 7 m (9.24 mm i.d.) plus 3 m (3.96 mm i.d.) long PFA sampling lines during PHOTONA 2

and 10 m long (9.24 mm i.d.) plus 2 m (3.96 mm i.d.) sampling lines during PHOTONA 3. The residence times in the sampling inlets were estimated to be between 1.6 and 3 s, which were short enough to avoid significant chemical conversions, e.g. by the reaction of $O_3$ with NO.

The different canopy heights used during the three campaigns required slight changes in the AG setup. During PHOTONA 1 and 2, the external sampling units of the LOPAPs were fixed on a small mast at heights of 0.15

and 1.5 m, and 0.3 and 1.5 m, respectively (see Figure 1). Note that the lowest height in PHOTONA 2 was just at the top of the canopy towards the end of the campaign, but was always above the displacement height (0.24 m, for definition see section 2.3). The canopy was also quite heterogeneous at that time as shown by the 15 % coefficient of variation of the canopy height and the area around the LOPAP had, on average, a lower canopy height. For PHOTONA 3, a scaffold tower of around 5.5 m in height with two levels was installed on the field,

on each of which a LOPAP was mounted (inlet sampling heights 3.0 m and 5.2 m). All other trace gases were measured at three heights during PHOTONA 1 (0.2, 0.7, 1.6 m) and 2 (0.4, 0.6, 1.5 m) and one height during PHOTONA 3 (5.0 m), using one instrument for each trace gas connected to Teflon solenoid valves (NResearch, USA). Measurements were made at 30 s intervals at all three heights (for details see Stella et al., 2012). During all campaigns the sampling inlets were positioned facing away from the field racks towards the prevailing wind

direction in order to minimize turbulence disruptions by the racks themselves. For EC measurements a sonic anemometer (R3, Gill Inc., UK) was mounted on a nearby mast at a height of 3.17 m during PHOTONA 1 and 2 and 5.0 m during PHOTONA 3.

Furthermore, meteorological parameters such as wind speed (*WS*) at different heights (cup anemometer, Cimel, FR), wind direction (*WD*) (W200P, Campbell Sci. Inc., USA), relative humidity (*RH*) and air temperature ($T_{air}$)

(HMP-45, Vaisala, FI) as well as soil parameters like the soil temperature ($T_{soil}$) (copper-constantan thermocouples) and soil water content (*SWC*) at different depths (TDR CS 616, Campbell Sci. Inc., USA) were measured continuously. The photolysis frequency *J(NO₂)* was measured using a filter radiometer (Meteorologie consult GmbH, Germany) during PHOTONA 1, 2 and 3 and a spectral radiometer (Meteorologie consult GmbH, Germany) during PHOTONA 3, by which also $J(O^1D)$ and *J(HONO)* were determined. During PHOTONA 1

and 2 *J(HONO)* was calculated from measured *J(NO₂)* using the method described by Kraus and Hofzumahaus (1998).

### 2.3. Aerodynamic gradient method

The HONO flux was calculated from the AG method by using a flux-profile relationship based on the Monin-Obukhov (MO) similarity theory that describes the non-dimensional gradient of a scalar $\chi$ (i.e. the concentration

of HONO, *c(HONO)*) as a universal function of the atmospheric stability parameter $(z - d) / L$ (e.g. Kaimal and Finnigan, 1994).

$$\frac{\kappa \cdot (z - d)}{\chi_*} \cdot \frac{\partial \chi}{\partial z} = \varphi_{(z-d)/L} \tag{1}$$

Here $\kappa$ is the von Karman constant (0.41), $\chi$ the measured scalar, $\chi_*$ the scaling parameter of $\chi$, $z$ the measurement height, $d$ the displacement height and $L$ the Obukhov length. The displacement height accounts for

the disturbance of the canopy on the flow, and was taken as $0.7 \cdot h_c$ ($h_c$: height of canopy), a common

parameterization in micrometeorology which was validated for this field site by Loubet et al. (2013). During the 60's and 70's of the last century a lot of effort was spent in the determination of the universal function $\varphi_{(z-d)/L}$ and its primitive $\Psi_{(z-d)/L}$ (Swinbank 1964; 1968; Businger et al., 1971). In the actual work, the universal function for heat $\Psi_{H,(z-d)/L}$ as published by Businger (1966) was integrated with the method of Dyer and Hicks (1970) for the

unstable case. For the stable case the universal function for heat as published by Webb (1970) was used (see Supplementary material).

The flux of a scalar, which is equal to $u_* \cdot \chi_*$, can be deduced from equation (1) which leads for HONO to (see supplementary material for the development of this equation):

$$F_{z_{ref}} = -\kappa \cdot u_* \cdot \frac{\partial c(HONO)}{\partial [\ln(z-d) - \Psi_{(z-d)/L}]} \tag{2}.$$

Here $c(HONO)$ is the concentration of HONO. $F_{z_{ref}}$ is representative of the flux at the geometric mean height of the concentration measurements, $z_{ref}$, which we hence define as $z_{ref} = \sqrt{(z_1-d) \cdot (z_2-d)}$ , where $z_1$ and $z_2$ are the measurement heights above the ground. The friction velocity $u_*$ and the Obukhov length $L$ were calculated from eddy covariance measurements as explained in detail in Loubet et al. (2011).

### 2.4. Data treatment

To interpret the flux data for each measurement campaign, a diurnal average was calculated by the formation of one-hour means from the whole measurement period. Using this procedure the errors of the individual measurements were reduced by averaging over a large number of values. However, some filtering steps were also applied which removed rain and high emission events from the data. These events led to higher noise in the daily patterns of the HONO flux and were therefore classified as unusual conditions or artefacts that did not

represent a common flux profile of the studied agricultural field site.

Finally, for the correlation analysis of the diurnal average (see section 3.4), weighted orthogonal regression fits (Brauers and Finlayson-Pitts, 1997) of the HONO flux against different variables were applied using the standard error (SE) of the one hour average for weighting (SE: standard deviation divided by the square root of $n$, the number of data). To assess the goodness of these fits the merit function $\chi^2$ and the goodness of fit parameter

Q were determined (Brauers and Finlayson-Pitts, 1997). A small $\chi^2$ and a large Q indicate a strong linear correlation of the analysed parameters.

### 2.5. Quality of the HONO flux

### 2.5.1. Estimation of the aerodynamic gradient uncertainty

The following main factors may influence the error of a flux calculation using the AG method. First of all, flux

gradient relationships have been studied for quite some time and show good similarities for trace gases such as $CO_2$ or sensible and latent heat using the above described universal functions, but there is always some uncertainty if using an indirect method. Moreover, for HONO the flux-similarity has never been compared to other techniques (e.g. the EC method). However, during PHOTONA 1 fluxes of nitric oxide (NO) and ozone ($O_3$) were measured additionally by eddy covariance (EC) and were in good agreement ($O_3$), or at least

comparable (NO) with fluxes calculated by the AG method (Stella et al., 2012). This demonstrated the applicability of the gradient method at the local homogeneous field site.





For the calculation of the uncertainties of the HONO flux by Eq. (2), errors of the gradient ($\sigma_{gradient}$) and of $u_*$ ($\sigma_{u*}$) are of direct importance. During all campaigns, HONO was measured at two heights using two LOPAPs. Hence, the quantification of the gradient strongly depended on the accuracy of these two instruments. The LOPAPs were therefore intercompared several times in the field, by placing the external sampling units beside each other and also by using a common PFA inlet line and a T-connection between the sampling inlets. In order to estimate the error of both instruments, again weighted orthogonal regressions (Brauers and Finlayson-Pitts, 1997) were applied, using the precision errors of both LOPAPs for weighting (see Figure 2). The inter-comparisons showed excellent agreement during PHOTONA 1 and 2, with a small intercept and a slope close to 1, demonstrating the capability of the used method to calculate gradients. Not quite so good agreement was obtained for PHOTONA 3, which may partly be explained by the lower HONO levels. To reduce systematic errors in the flux calculation, the lower LOPAP was harmonized using the linear regression fits shown in Figure 2.

The error of the gradient was then calculated from the precision of the instruments ($\sigma_{LOPAP}$) and the errors of the slope ($\Delta b$) and the intercept ($\Delta a$) of the regression fit (see Figure 2), using 95 % confidence intervals ($2\sigma$). The HONO concentration, $c(HONO)$, always refers to the higher value of both instruments in order to obtain the maximum deviation.

$$\sigma_{gradient} = \sqrt{\sigma_{LOPAP}{}^2 + \Delta b^2 \cdot c(HONO)^2 + \Delta a^2} \qquad (3)$$

The uncertainty of the flux during PHOTONA 1 was finally calculated by error propagation using $\sigma_{gradient}$ and $\sigma_{u*}$ (for further details of the calculation of $\sigma_{u*}$, see Stella et al., 2012). For PHOTONA 2 $\sigma_{u*}$ was calculated from 5 min data of $u_*$ ($n = 6$). For PHOTONA 3 the uncertainty of the flux was not calculated, as only $\sigma_{gradient}$ was available.

### 2.5.2. Influence of the roughness sub-layer

The flux-gradient-similarity is not valid inside the roughness sub-layer (RSL), which ranges from the canopy top to around two times the canopy height (Cellier and Brunet, 1992). In the present study, the flux divergence caused by the RSL was analysed using the methods of Cellier and Brunet (1992) and Graefe (2004). However, the influence of the RSL during both canopy campaigns was only of minor importance and therefore neglected for further interpretation of the flux data.

### 2.5.3. Dealing with chemical reactions in the gas phase

The aerodynamic gradient method is strictly valid only for non-reactive trace gases. However in the present study, the photolysis and the production of HONO (e.g. by NO+OH) in the gas phase below the measurement heights may create artificial fluxes that need to be corrected for.

To check for chemical reactions during turbulent transport the so-called Damköhler number ($Da$) has been used: $Da = \tau_{trans} / \tau_{chem}$. It compares the chemical reaction time scale ($\tau_{chem}$, see Eq. (4)) with the transport time scale ($\tau_{transp}$, see Eq. (5)) to identify periods when chemical reaction may generate flux divergence. To calculate the chemical time scale of HONO only its photolysis was taken into account, which is the dominant destruction path of HONO during daytime:

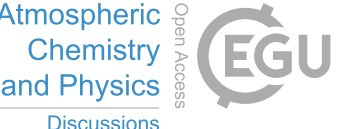



$$\tau_{chem} = \frac{1}{J(HONO)} \qquad (4)$$

In contrast, loss or production rates by the reactions HONO + OH and NO + OH are typically more than an order

of magnitude lower than the HONO photolysis even when considering a typical maximum OH concentration of

$5 \cdot 10^6$ cm$^{-3}$ during daytime. For the correction of chemistry, the transport time scale depends on a) the location of

the HONO source and b) the region where HONO chemistry starts to become important. Since photolysis is

diminished in the canopy due to shadowing by the leaves, we only consider the transfer time between the canopy

exchange height (defined as $d + z_0$') and the reference height ($z_{ref}$). This leads to the following definition of $\tau_{trans}$:

$$\tau_{trans} = \frac{1}{R_a \cdot (z_{ref} - d - z_0) + R_b \cdot (z_0 - z_0')} \qquad (5)$$

Where $R_a$ is the aerodynamic resistance for transfer between $d + z_0$ ($z_0$ is the roughness height for momentum)

and $z_{ref}$. $R_b$ is the canopy boundary layer resistance for HONO accounting for the transfer between the roughness

height $d + z_0$ and the canopy exchange height located at $d + z_0'$ ($z_0'$ is the roughness height for the scalar). $R_a$ and

$R_b$ were estimated by:

$$R_a = \frac{u_{z_{ref}}}{(u_*)^2} \qquad (6)$$

$$R_b = \frac{1.45 \cdot Re^{0.24} \cdot Sc^{0.8}}{u_*} \qquad (7)$$

Here $u_{z_{ref}}$ is the wind speed at $z_{ref}$, $Re$ is the canopy Reynolds number, $Re = u_* \cdot z_0 / \nu_a$, and $Sc$ the Schmidt

number, $Sc = \nu_a / D(HONO)$, where $\nu_a$ is the kinematic viscosity of air and $D(HONO)$ the diffusivity of HONO

in air (Garland et al., 1977). During PHOTONA 3, for which the direct measured photolysis rate $J(HONO)$ was

available, the transport time during daytime was typically of the order of a minute and much smaller than the

chemical lifetime of HONO of $\tau_{chem} \geq 10$ min. Thus, the influence of the photolytic loss to the overall HONO

flux was always below 10 % ($Da < 0.1$) and we considered a refinement of the analysis by the stability

corrections on $R_a$ (see Stella et al., 2012) of less importance. As no production terms for HONO were considered

for the calculation of the flux divergence, the influence of photolysis gives even only an upper limit for the flux

divergence. Similar results were obtained for PHOTONA 1 and 2 using calculated $J(HONO)$ data. For further

analysis, errors by chemical reactions were neglected, which will, however, not significantly influence the

interpretation of potential precursors and driving forces of the HONO flux.

### 2.5.4. Footprint area

The field site in Grignon is quite homogenous although with a slight slope and some building and trees around

600 m to the west. To decide if the flux is influenced by surfaces outside this area that may disturb homogeneity,

a footprint analysis, as described by Neftel et al. (2008), has been performed using the model ART Footprint

Tool version 1.0, which is available from http://www.agroscope.admin.ch/art-footprint-tool. The influence of the

field site was >92 % (median of all campaigns) and very comparable to other flux measurements at this location.

For example, Loubet et al. (2011) stated that up to 93 % of the field was inside the mast footprint (3.17 m height)

during summer-spring campaigns.





## 3 Results and discussion

### 3.1. General observations

Main standard meteorological measurements and mixing ratios from all campaigns are presented in Figure 3. During PHOTONA 1, maximum daytime temperatures ranged from 21 to 28 °C and daytime relative humidities were around 30 to 40 %. Minimum night time temperatures ranged from 10 to 17 °C with relative humidities between 60 and 95 %. Dry conditions generally prevailed with only one moderate rain event on the 24$^{th}$ August (1.2 mm h$^{-1}$). During most of the campaign the wind came from the south-west with only a short period of two days dominated by north-easterly winds starting on 22$^{nd}$ August. Maximum wind speeds during the day varied from 2 to 8 m s$^{-1}$, with friction velocities up to 0.46 m s$^{-1}$. Minimum HONO mixing ratios during the day varied from 5 to 120 pptV with maximum morning peaks of up to 700 pptV. Minimum daytime mixing ratios of NO$_2$ were around 1 ppbV with maximum mixing ratios during the morning hours of up to 25 ppbV. Nocturnal mixing ratios of NO$_2$ varied typically from <1 ppbV, in the middle of the night and up to 25 ppbV in the late evening.

Maximum daytime temperatures during PHOTONA 2 varied from 10 to 19° C with relative humidities in the range 31–65 %. Minimum nocturnal temperatures, reached before early morning, ranged from 2.4 to 7.7 °C with relative humidities between 76 and 93 %. Moderate rain events (up to 1.2 mm h$^{-1}$) occurred during the beginning (8$^{th}$ April) and in the middle (13$^{th}$ April) of the campaign. Maximum wind speeds during the day varied from 3 to 9 m s$^{-1}$, i.e. comparable to PHOTONA 1. However, the canopy generated turbulences, which is expressed in higher friction velocities with maximum values up to 0.6 m s$^{-1}$. HONO mixing ratios during early afternoon varied from 60 to 450 pptV and reached up to 900 pptV during the night. For NO$_2$ one short emission event with air mass coming from the Paris region occurred during the first day with mixing ratios of up to 60 ppbV. Thereafter, the mixing ratios varied from <20 ppbV during morning hours to around 1 ppbv in the afternoon.

For the first half of the PHOTONA 3 campaign, maximum daytime temperatures reached values up to 31 °C, but decreased to around 16° C during the second half of the campaign. A similar trend was observed for the nocturnal temperatures with minimum values in the range 16 to 22 °C in the beginning and 6 to 12 °C at the end of the campaign. Minimum relative humidities in the afternoon ranged from 40 to 80 % and maximum humidities during early morning were in the range 80 to 97 %. Light to strong rain events (0.1-8 mm h$^{-1}$) occurred on the 21$^{st}$, 22$^{nd}$, 26$^{th}$ and 27$^{th}$ of August and led to decreases in the HONO mixing ratios, possibly caused by the high effective solubility of HONO in water. Friction velocities reached values up to 0.6 m s$^{-1}$ and were comparable to those of PHOTONA 2. HONO mixing ratios reached values up to 600 pptV during the early morning and decreased to around 20 pptV in the afternoon. Maximum NO$_2$ mixing ratios with values up to 30 ppbV were reached in the night or during early morning hours and minimum mixing ratios down to <1 ppbV were observed in the afternoon.

### 3.2. Diurnal average HONO flux

The HONO flux showed similar profiles in the summer campaigns PHOTONA 1 and 3 but was different in the spring campaign PHOTONA 2 (see Figure 4). Minimum emissions, or even depositions (PHOTONA 2) occurred at night and emissions were observed during daytime with a morning peak at around 8:00 UTC (Coordinated Universal Time). During daytime of the summer campaigns (PHOTONA 1 and 3) continuously decreasing HONO fluxes were observed after the morning peak, whereas during the spring campaign the flux rapidly decreased after a strong morning peak and stayed more or less constant throughout the rest of the day.





For PHOTONA 1 and 3 the HONO flux then decreased again to a minimum around midnight or slightly earlier. The magnitudes of the observed daytime fluxes in the range 0.1 to 2.3 ng N $m^{-2}$ $s^{-1}$ (0.05-1·$10^{14}$ molec. $m^{-2}$ $s^{-1}$, see Figure 4) are comparable to measurements of other studies in suburban/rural regions. Ren et al. (2011), for example, found fluxes during daytime with a maximum around 1.4 ng N $m^{-2}$ $s^{-1}$ on average during CalNex 2010 in California and Zhou et al. (2011) obtained maximum daytime HONO fluxes during noon and early afternoon

of around 2.7 ng N $m^{-2}$ $s^{-1}$ on average on the PROPHET tower in Michigan. The observed morning peak is also in agreement with another study, where the measurements were performed in and above a forest canopy (He et al., 2006) and were explained by dew evaporation.

The range of daytime HONO fluxes measured in this study is also of the order of magnitude of the laboratory derived "optimum HONO emission flux" by biological processes for the soil collected from the Grignon field

site, which was 6.9 ng N $m^{-2}$ $s^{-1}$ (Oswald et al., 2013). In these experiments optimum HONO emissions were derived during drying of the soil surface in a dark chamber by flushing with completely dry air. The cited maximum emission for the Grignon soil was obtained at a soil temperature of 25°C and at a low soil humidity of around 10 % of the water holding capacity (*whc*), which corresponds to a gravimetric soil water content of 5.5 % (*whc* = 54.9% in gravimetric humidity). Multiplying by the soil density at the surface (1.3±0.5 kg $L^{-1}$), gives the

corresponding soil water volume content of 7.1 % much lower than those at the present field site, where soil water contents and soil temperatures at 5 cm depth of 13.2±0.4 % and 22.6±9.7 °C in PHOTONA 1, 27.1±2.0 % and 10.1±4.2 °C in PHOTONA 2 and 27.7±1 % and 18.4±4.1 °C in PHOTONA 3 were observed. According to the soil humidity and temperature response curves reported by Oswald et al. (2013), biological emissions of HONO are expected to be lower than 5 ng N $m^{-2}$ $s^{-1}$ in PHOTONA 1, and lower than 0.001 ng N $m^{-2}$ $s^{-1}$ in

PHOTONA 2 and PHOTONA 3. Hence we expect the biological source as evaluated by Oswald et al. (2013) to be negligible in PHOTONA 2 and 3, while it could be comparable to the measured HONO flux in PHOTONA 1.

### 3.3. Correlation between fluxes and concentrations of HONO

When plotting the night-time data of the HONO flux against the HONO concentration for PHOTONA 2 (and to lesser extent for PHOTONA 1) a significant positive correlation is observed (PHOTONA 2: $R^2$ = 0.92;

PHOTONA 1: $R^2$ = 0.43) with negative HONO fluxes at HONO mixing ratios <0.43 ppb and <0.13 ppb for PHOTONA 2 and 1, respectively. This observation indicates a significant impact of HONO deposition on the net HONO flux and is in agreement with the observed negative net HONO fluxes observed in the early morning of PHOTONA 2 (see Figure 4). In contrast, for the night-time data of PHOTONA 3 and the daytime data of all three campaigns there was no significant correlation between the HONO flux and its concentration. The missing

daytime correlation supports that deposition is of minor importance compared to the more important HONO source terms during daytime.

### 3.4. Correlation between fluxes of HONO and potential precursors

As the major aim of the present study was to explain the origin of HONO sources during daytime, the following section concentrates on the flux and its correlations with potential precursors using only data from 6:00 to 20:00

UTC. Campaign averaged HONO fluxes (see Figure 4) were plotted against different potential precursors and controlling parameters. Correlations of the HONO flux with the product of the photolysis frequency and concentration of $NO_2$, *J(NO_2)·c(NO_2)*, were observed for all campaigns (see Table 1). Especially the HONO





fluxes during PHOTONA 1 and 3 were well explained by $NO_2$ and UV-A light intensity expressed by $J(NO_2)$, which is presented exemplarily for PHOTONA 1 in Figure 5.

While a correlation between the daytime HONO flux and the product of $J(NO_2) \cdot c(NO_2)$ was observed for all three campaigns, especially during the two summer campaigns PHOTONA 1 and 3, an additional correlation with the friction velocity was observed during the spring campaign PHOTONA 2, see Table 1. Reasons for the different correlation results and the different diurnal shapes of the HONO fluxes between the two summer and the spring campaigns (see Figure 4) are still not fully clear. A potential explanation could be the higher influence

of HONO deposition during the colder spring campaign (see below) masking the correlation with the main proposed source precursors $NO_2$ and radiation. Since deposition fluxes will depend on the turbulent vertical mixing this could explain the higher correlation with the friction velocity. Alternatively, decoupling between the regimes above and below a dense canopy will also depend on the vertical turbulent mixing (Sörgel et al., 2011a) and may have influenced the HONO flux from the soil source region to the measurements heights above the

canopy. Finally, stomatal uptake of HONO by the leaves of the triticale canopy, especially during daytime (Schimang et al., 2005), may have caused the lower daytime fluxes during PHOTONA 2 (see Figure 4) compared to the other campaigns.

The finding of a light and $NO_2$ dependent HONO flux is in good agreement with the study of Ren et al. (2011), where daytime HONO fluxes above an agricultural field also correlated well with the product of $NO_2$

concentrations and incident solar radiation during the CalNex 2010 campaign. Also the very weak correlation of the HONO flux with the $NO_2$ concentration above a forest canopy at the PROPHET site (Zhou et al., 2011; Zhang et al., 2012) can be attributed to an influence of the canopy. Correlations of HONO with its precursors are expected to become worse when measurements are carried out above high trees as at the PROPHET site, which are able to fully decouple the ground surface from the air above the canopy (Sörgel et al., 2011a; Foken et al.,

2012). The results from the present study are qualitatively also in good agreement with former studies in which the daytime source of HONO was quantified using the PSS approach and in which also a strong correlation of the daytime source with radiation and/or $NO_2$ was observed (Elshorbany et al., 2009; Sörgel et al. 2011b; Villena et al., 2011; Wong et al., 2012; Lee et al., 2016; Meusel et al., 2016). This observation may imply a mechanism of HONO formation by the reduction of $NO_2$ with organic photosensitizer materials like humic acids as proposed

in laboratory studies (George et al., 2005; Stemmler et al., 2006; 2007; Han et al., 2016).

Another HONO source, microbiological formation of nitrite in the soil, as proposed by Su et al. (2011) and Oswald et al. (2013), should strongly depend on the soil temperature and the soil surface water content, due to the temperature dependence of the solubility of HONO in soil water and/or the adsorption of HONO on the soil surface and the biological activity of the soil. Here, the HONO fluxes are expected to increase with increasing

temperature and decreasing humidity. However, except for PHOTONA 2 the correlations of the HONO flux were much weaker with the soil temperature compared to those with $J(NO_2)$ and with the product $J(NO_2) \cdot c(NO_2)$ (see Table 1). In addition, the HONO fluxes showed no significant correlation with the soil water content, the relative humidity of the air or its inverse. Also based on the observed diurnal shape of the HONO flux, microbiological formation of nitrite/HONO on the soil surfaces seems to be unlikely, since the highest fluxes

would be expected at low soil water content and high temperature, leading to a maximum of the HONO flux in the early afternoon, when the soil surface is at its driest and warmest due to irradiation from the sun. In contrast, the highest fluxes were observed during the morning in the present study (see Figure 4). And finally, the





expected optimum HONO fluxes were much lower in PHOTONA 3 compared to PHOTONA 1 due mainly to the different soil water contents (see section 3.2), while the measured fluxes were very comparable (see Figure

4). Thus, although the laboratory derived optimum HONO fluxes were in the same range as those observed in the present field study during PHOTONA 1 (see section 3.2), the different diurnal shapes and seasonal variability of the expected and measured HONO fluxes do not support the microbiological soil mechanism proposed by Su et al. (2011) and Oswald et al. (2013) as a major HONO source at the present field site. This result is in good agreement with another recent field study in which the daytime HONO source could also not be

explained by a biological soil source, but showed a strong correlation with the radiation (Oswald et al., 2015), similar to that observed in the present study. It should be stressed that in the Oswald et al. (2013) study the experimental conditions were not representative for the present field site. While in these laboratory studies the upper soil surface was flushed by completely dry air, leading to optimum HONO emissions only at very dry conditions, the relative humidity never decreased below 26 %, 31 % and 43% in PHOTONA 1, 2 and 3,

respectively. More work is desirable to reconcile HONO field data with incubation experiments as performed by Oswald et al. (2013). Finally, we can not completely exclude this source here, as we observed a small positive intercept in the correlation plots of the HONO flux against $J(NO_2) \cdot c(NO_2)$ in all campaigns (cf. Figure 5 for PHOTONA 1). Since the biological soil source is expected to be light- and $NO_2$-independent (Su et al., 2011; Oswald et al., 2013) this intercept may reflect the magnitude of this source and/or other light-independent

sources. However, the small magnitude of the intercept compared to the observed HONO fluxes, especially for PHOTONA 1 and 3, suggests that light-independent sources are of minor importance during daytime.

The lack of information about nitrate surface concentrations during the present study does not allow us to directly exclude a $HNO_3$ photolysis mechanism as proposed by Zhou et al. (2011), who observed a HONO flux that is positively correlated with leaf surface nitrate loading and light intensity. However in the present study, a

better correlation of the HONO flux with $J(NO_2)$ (near UV-A) of $R^2 = 0.38$ than with $J(O^1D)$ (UV-B) of $R^2 = 0.17$ was observed for PHOTONA 3 for which a spectroradiometer was used to measure both photolysis frequencies (see Table 1). Since $HNO_3$ photolysis is expected to be active mainly under short wavelength UV radiation, while the photosensitized conversion of $NO_2$ on humic acid surfaces works well already in the visible and near UV-A (Stemmler et al., 2006; Han et al., 2016), the latter mechanism seems to be a more likely HONO

source at the present field site compared to photolysis of adsorbed $HNO_3$. This is confirmed by the high correlation of $F(HONO)$ with the product $J(NO_2) \cdot c(NO_2)$ of $R^2 = 0.85$ (see Table 1). In addition, for a potential nitrate photolysis source a maximum of the HONO flux would be expected in the afternoon due to a number of contributing factors, i.e. (i) the main $HNO_3$ source during daytime is the reaction of $NO_2$ with OH, (ii) the typical diurnal profiles of the OH concentration and (iii) the subsequent deposition of gas-phase $HNO_3$ onto ground

surfaces. In contrast, the campaign averaged HONO fluxes showed asymmetric diurnal profiles with a maximum in the morning, which can be explained by the higher $NO_2$ morning levels compared to the afternoon (see Figure 4). Finally, in a recent laboratory study on the photolysis of adsorbed $HNO_3$ only a very low upper limit photolysis frequency of $J(HNO_3 \rightarrow HONO) = 2.4 \cdot 10^{-7} \, s^{-1}$ (0° SZA, 50 % r.h.) was determined (Laufs and Kleffmann, 2016), which is too low to explain any significant HONO formation in the atmosphere.

Another recently discussed mechanism, the acid displacement of HONO by deposition of strong acids (e.g. VandenBoer et al., 2015), also seems to be unlikely for the present field site. This proposed source should maximize in the afternoon because of the daytime formation of the main strong acid $HNO_3$ und its subsequent





deposition on ground surfaces (see discussion above and see Figure 4c in VandenBoer et al., 2015). In contrast, for any $NO_2$ dependent photochemical source a maximum HONO flux during morning hours is expected (see

Ren et al., 2011 and Figure 4c in VandenBoer et al., 2015) since the highest $NO_2$ concentrations occur in the morning and not in the afternoon (cf. Figure 4 for the present study). Only if the majority of the soil acidity results from night-time dry deposition of $N_2O_5$, the higher morning fluxes of HONO might be explained by the acid displacement mechanism. Here flux measurements of $HNO_3$ and $N_2O_5$ are necessary in the future. However, since we expect a higher contribution of $HNO_3$ uptake to the soil acidity, the asymmetric HONO flux profile

with higher values in the morning indicates that the acid displacement is of less significance for the present field site (and also for the data shown in Figure 4c in VandenBoer et al., 2015).

### 3.5. Comparison of all campaigns

In order to find parameters that control the HONO flux in a kind of manner that is not visible using the individual campaign data, we tried to find parameters that affect the HONO flux using the data from all three

campaigns. Figure 6 shows HONO fluxes during PHOTONA 1, 2 and 3 as a function of the soil temperature. Although HONO fluxes of the individual campaigns correlated better with $J(NO_2) \cdot c(NO_2)$, see Table 1, an additional positive correlation of all the data with the soil temperature is obvious. With increasing soil temperature the net HONO flux increases, which may be explained by a temperature dependent adsorption/solubility process (Su et al., 2011), which becomes more important at lower temperatures compared

to the HONO source reactions. In the present study, only net HONO fluxes were quantified, which are controlled by typically smaller negative deposition fluxes and stronger positive formation by heterogeneous processes on the soil surface. When plotting the logarithm of the positive HONO fluxes against the inverse temperature a formal activation enthalpy for HONO formation of 41.2 kJ mol$^{-1}$ can be derived (see Figure 6). Assuming that HONO formation by $NO_2$ conversion on the soil surface is controlled by the temperature dependent HONO

solubility in the soil water, this activation enthalpy is in good agreement with the value of the enthalpy of solvation of HONO in water of $\Delta_{sol}H$ = -40.5 kJ mol$^{-1}$ (Park and Lee, 1988). The different signs of the two enthalpies are explained by the different reference points describing the same process, for which increasing solubility leads to a decrease in the HONO flux.

In conclusion, positive daytime HONO fluxes are explained in the present study by a $NO_2$ and light dependent

source, i.e. by the photosensitized conversion of $NO_2$ on soil surfaces (Stemmler et al., 2006) additionally controlled by the temperature dependent HONO adsorption on the soil or its solubility in soil water.

### 3.6. Parameterization of the HONO flux

The above results of the above correlation study were used to set-up a simple parameterization that describes the HONO flux for all campaigns. As the strongest correlation was observed for the HONO flux with the product of

$NO_2$ concentration with light intensity, a proposed photo-sensitized HONO source (Stemmler et al., 2006) was parameterized by the term $A \cdot J(NO_2) \cdot c(NO_2)$, see equation (8). To also describe the night-time HONO flux, which would have been zero when considering only this light-dependent source, an additional slower dark formation of HONO by heterogeneous $NO_2$ conversion on soil surfaces (e.g. Finlayson-Pitts et al., 2003 or Arens et al., 2002) was introduced by using a second source term $B \cdot c(NO_2)$. Because of the observed temperature





dependence of the HONO fluxes, both sources were multiplied by a Boltzmann term, for which the negative value of the experimental solvation/adsorption enthalpy of HONO of -41.2 kJ mol⁻¹ (see Figure 6) was used.

The two proposed sources are in agreement with results from several field and laboratory studies (Kleffmann, 2007), but would result in only positive modelled HONO fluxes. However, during PHOTONA 2 also net HONO deposition was observed in the early morning at the low soil temperatures of the spring campaign (see Figure 4).

To describe this net HONO uptake on ground surfaces an additional temperature dependent HONO deposition term was included, i.e. the product of the HONO concentration measured at the lower sampling height with a temperature dependent deposition velocity, $v(HONO)_T$. Finally, since the magnitude of HONO sources and sinks are expected to positively correlate with humidity (Finlayson-Pitts et al., 2003; Stemmler et al., 2006; Han et al., 2016; Su et al., 2011), all variables were optimized for a reference relative humidity ($RH$) of 50 % and were

scaled linearly with humidity ($RH/50\%$), leading to the final equation (8):

$$F(HONO)_{mod} = \left[ \left( A \cdot J(NO_2) \cdot c(NO_2) + B \cdot c(NO_2) \right) \cdot exp\left( \frac{\Delta_{sol}H}{R \cdot T_{soil}} \right) - c(HONO) \cdot v(HONO)_T \right] \cdot \frac{RH}{50\%} \quad (8)$$

The constants $A$ and $B$ were adjusted to obtain (i) a slope of one, (ii) an intercept of zero and (iii) a high correlation between modelled and measured HONO fluxes ($R^2 = 0.68$), resulting in final values for $A$ and $B$ of $2.9 \cdot 10^6$ m and $2.0 \cdot 10^{-3}$ m s⁻¹, respectively. When considering also for the Boltzmann and humidity terms, the

final value for $A$ is in good agreement with the average experimental value of $2.5 \cdot 10^6$ m determined from the correlation plots of the three campaigns (see section 3.3.2). This indicates that the photolytic HONO source is mainly controlling the net HONO fluxes during daytime. The second term $B$ multiplied by the Boltzmann and humidity terms can be described as the effective deposition velocity of $NO_2$ to form HONO in the dark on ground surfaces. The measured overall deposition velocity of $NO_2$ during PHOTONA 1 varied from 0.002 m s⁻¹

during night-time to 0.0055 m s⁻¹ before noon (calculated from the diurnal average data of the whole campaign, see Stella et al., 2011). Dividing the effective deposition velocity for HONO formation in the dark ($B$ multiplied by the Boltzmann and humidity terms) by the overall measured deposition velocity of $NO_2$, resulted in campaign averaged ratios in the range 2.0 % (day) to 4.4 % (night), i.e. only 2–4.4 % of the $NO_2$ uptake on ground surfaces leads to HONO production by the heterogeneous dark conversion of $NO_2$. This range of values is comparable

with nigh-time observations of Stutz et al. (2002), who calculated a conversion efficiency to form HONO from $NO_2$ deposition of 3±1 %.

When comparing the two proposed sources, the dark conversion of $NO_2$ contributed only ~10 % to the HONO fluxes around noon, while it was the only source during night-time by definition. When integrating over the whole day (24 h), the dark conversion contributed 23 %, 28 % and 30 % to the total heterogeneous HONO

production, while the photochemical source was 3.3, 2.6 and 2.3 times larger during PHOTONA 1, 2 and 3, respectively. These results are in general agreement with former field studies using the more simple PSS approach in which the photochemical HONO source also dominates daytime production (Kleffmann, 2007 and references therein), while the dark conversion of $NO_2$ is controlling the night-time build-up of HONO and the OH radical production in the early morning after sunrise (e.g. Alicke et al., 2002).

To describe also the negative HONO fluxes during the PHOTONA 2 spring campaign (see Figure 4), the temperature dependent effective HONO deposition velocity ($v(HONO)_T$, see equation (8)) was adjusted to values of 0.02 m s⁻¹ at 0°C decreasing exponentially to non-significant values at 40 °C ($v(HONO)_T$ = exp(23920/T-



91.5). The higher end deposition velocity used here is in agreement with published upper limit values in the range 0.005 m s$^{-1}$ (Stutz et al., 2002), 0.017 m s$^{-1}$ (Harrison and Kitto, 1994; Trebs et al., 2006) and 0.06 m s$^{-1}$

(Harrison et al., 1996). Based on this model adjustment, HONO deposition became more significant towards the end of the night, especially during PHOTONA 2, when modelled deposition fluxes were up to four times larger compared to the sources. In contrast, during daytime, deposition fluxes were less significant and made up only a few percent at maximum compared to the source reactions, in agreement with the missing correlation of the HONO flux with its concentration during daytime (see section 3.3).

The measured HONO fluxes were well described by equation (8) especially during PHOTONA 1 and 3, see Figure 7. However, during PHOTONA 2, the campaign with the triticale canopy, the daytime HONO fluxes were overestimated by the model, which may be explained by additional stomatal uptake of HONO by the leaves (Schimang et al., 2005) during transport of HONO from the proposed soil surface source region to the sampling positions above the dense canopy. In addition, the sharp measured morning peak of the HONO flux during

PHOTONA 2 is also not well represented by the model. This morning peak may be explained by dew evaporation of accumulated nitrite (formed, e.g., by dark reactions of NO$_2$ or uptake of HONO) from vegetation surfaces when the temperature increased in the morning, which is in agreement with results from other field studies (Rubio et al., 2002; He et al., 2006).

## 4 Conclusion

The present study demonstrates the useful application of the aerodynamic gradient method together with the LOPAP technique to measure HONO fluxes over bare soil and canopy surfaces.

Correlation studies of the HONO flux point towards a light driven HONO source during daytime fed by NO$_2$, which is in line with a photosensitized reaction of NO$_2$, e.g. on humic acid surfaces as observed in laboratory studies. In addition, the comparison of the three campaigns shows an additional influence of the soil temperature

on the HONO flux suggesting that adsorption of HONO on the soil surface is of additional importance.

A simple model using two NO$_2$, temperature and humidity dependent HONO source terms and a temperature dependent HONO adsorption was able to reproduce quite satisfactory the measured HONO fluxes, at least for the two PHOTONA summer campaigns. In agreement with known sources of HONO observed in laboratory studies, HONO formation by heterogeneous conversion of NO$_2$ on ground surfaces is proposed via (i) a slower

reaction in the dark and (ii) a faster photosensitized reaction scaling with $J(NO_2)$. The photosensitized source (ii) accounted for ca. 90 % of the daytime HONO formation and was still ca. three times stronger compared to (i) when integrated over the whole day in excellent agreement with former field studies using the simpler PSS approach.

**Acknowledgement**

The support of this work by the Deutsche Forschungsgemeinschaft (DFG) under contract number (KL 1392/3-1) is gratefully acknowledged. The three experimental campaigns were conducted in the fluxnet Fr-GRI site supported by INRA. This work was also supported by the French LEFE-CNRS-INSU and R2DS (CNRS, Région Ile de France) programs and ANR project Vulnoz (ANR-08-VULN-012). The experimental campaigns were also supported by European FP7-NitroEurope (project 017841), FP7-Eclaire (FP7-ENV-2011-282910) and ICOS




projects. The authors also gratefully acknowledge Bernard Defranssu, Dominique Tristan and Jean-Pierre de Saint-Stéban from the experimental farm of AgroParisTech Grignon providing access to their fields, Michel Burban, Brigite Durand, Olivier Fanucci, Nicolas Mascher and Jean-Christophe Gueudet for their support in the field.

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





Table 1: Goodness of the weighted orthogonal regressions of hourly average daytime data (6:00 to 20:00 UTC) of *F(HONO)* against different variables for the three PHOTONA campaigns. The numbers represent $\chi^2/Q$ ($R^2$) values for which lower $\chi^2$ and higher $Q$ and $R^2$ values indicate better correlations (for definition see Brauers and Finlayson-Pitts, 1997). Bold numbers represent the strongest correlations observed for each campaign.

| | *J(NO₂)* | *J(O¹D)* | *T_soil* | *u*∗* | *J(NO₂)·c(NO₂)* |
|---|---|---|---|---|---|
| **PHOTONA 1** | 27.5/0.004 (0.47) | not measured | 50.2/6·10⁻⁷ (0.22) | 23.4/0.016 (0.41) | **7.27/0.78 (0.79)** |
| **PHOTONA 2** | 12.1/0.28 (0.27) | not measured | 9.33/0.50 (0.019) | **5.66/0.84 (0.37)** | 12.4/0.26 **(0.37)** |
| **PHOTONA 3** | 53.7/3·10⁻⁷ (0.38) | 79.8/5·10⁻¹² (0.17) | 121/5·10⁻²⁰ (0.03) | 62.7/7·10⁻⁹ (0.20) | **3.26/0.994 (0.85)** |

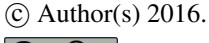



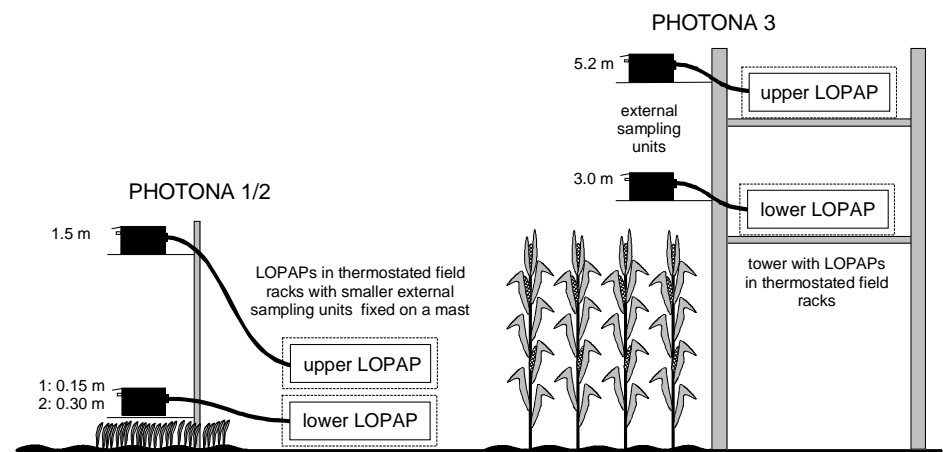


**Figure 1: HONO aerodynamic gradient setup during the PHOTONA campaigns. Left: PHOTONA 1 (bare soil) and 2 (triticale canopy) with the two external sample units fixed on a mast. Right: Scaffold tower during PHOTONA 3 (maize canopy) with the LOPAPs placed at different levels on the tower.**






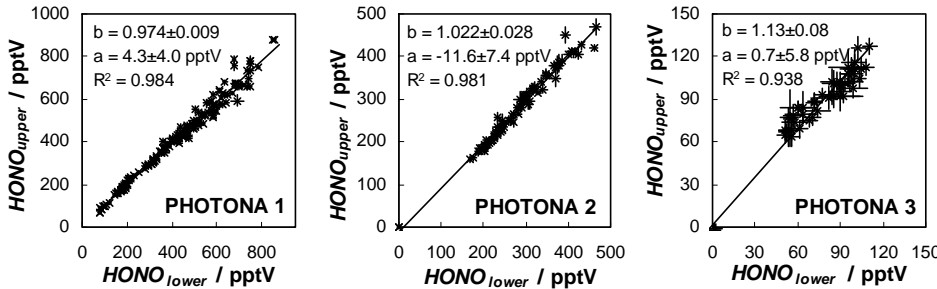

**Figure 2: HONO mixing ratios measured at the same height during PHOTONA 1-3 (from left to right). The solid lines show linear weighted orthogonal regressions (Brauers and Finlayson-Pitts, 1997) between the two instruments. The slope (b) and intercept (a) are given with their (2σ) standard deviations.**




**Figure 3:** Time series of mixing ratios of the main species HONO, NO$_2$, NO and O$_3$, meteorological parameters and *J(NO$_2$)* during the PHOTONA project (left to right: PHOTONA 1, 2 and 3).






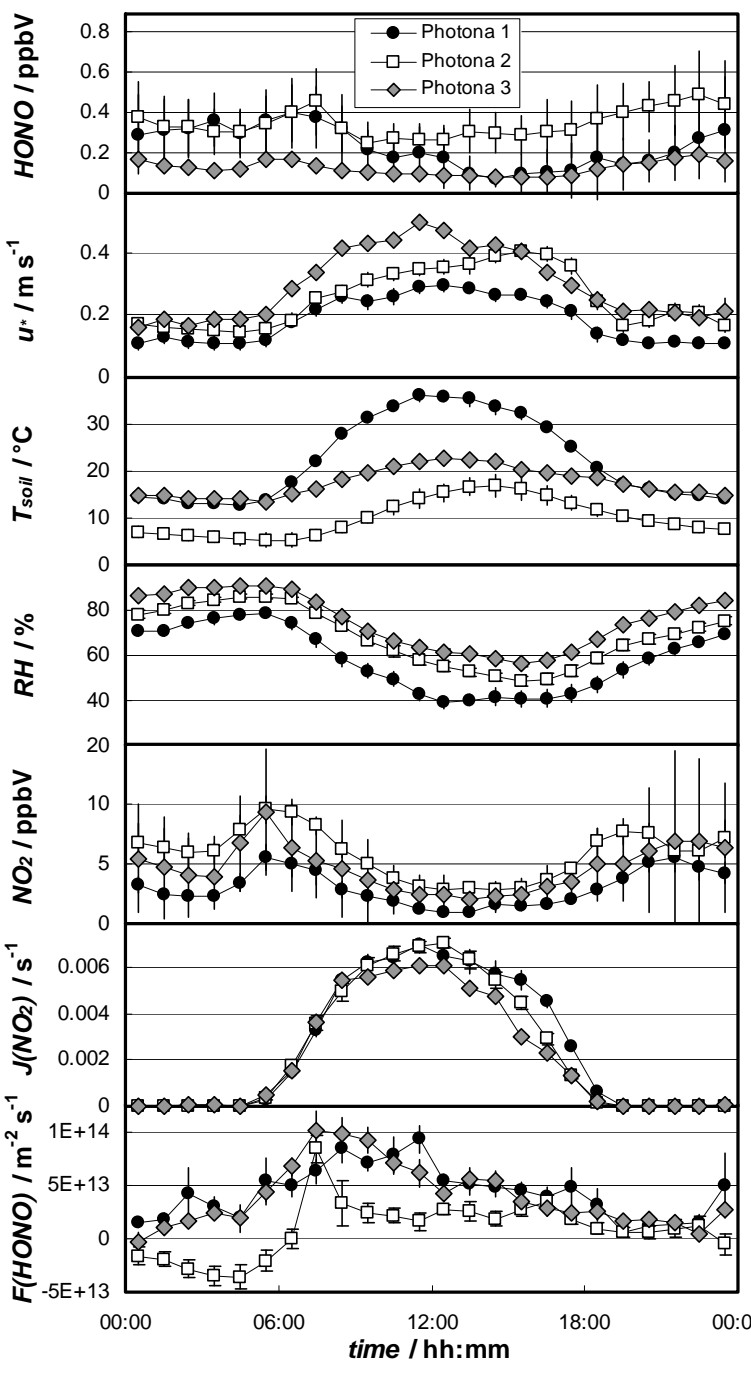

**Figure 4: Diurnal average data of the HONO flux, *F(HONO)*, and its potential precursors and driving factors *HONO*, *u∗*, *T_soil*, *c(NO₂)* and *J(NO₂)* during the three PHOTONA campaigns.**

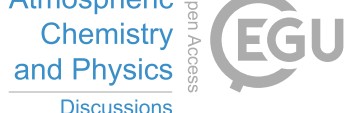




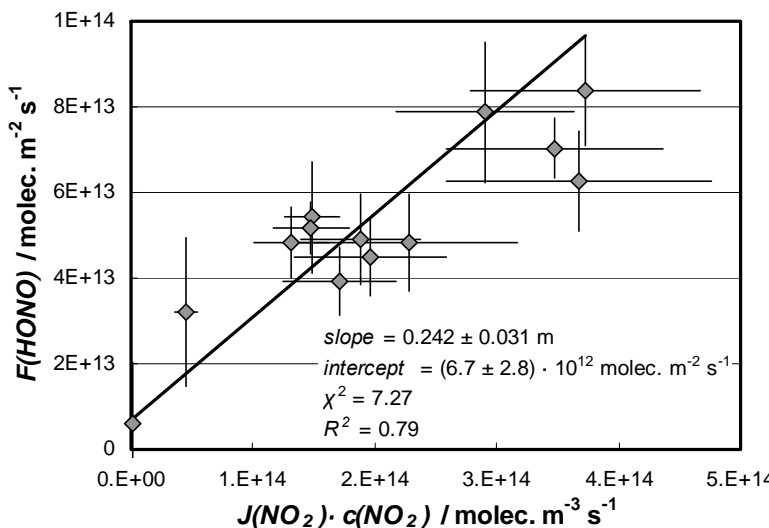

**Figure 5: Correlation of the diurnal HONO flux (6:00 to 20:00 UTC) with the product $J(NO_2)\cdot c(NO_2)$ during PHOTONA 1 with a weighted orthogonal regression fit (Brauers and Finlayson-Pitts, 1997).**




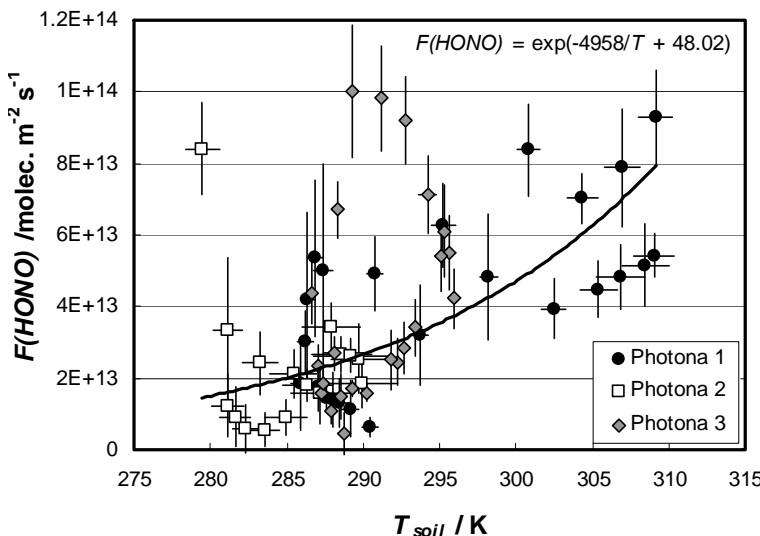

**Figure 6: Average diurnal HONO fluxes of all individual campaigns as a function of the soil temperature. The black line presents the regression fit using the exponential function: $F(HONO) = \exp(\Delta_{sol}H)/RT_{soil}+C)$, with $R$: universal gas constant (8.314 J mol$^1$ K$^{-1}$), $\Delta_{sol}H$: experimental enthalpy of solvation (-41.2 kJ mol$^{-1}$).**






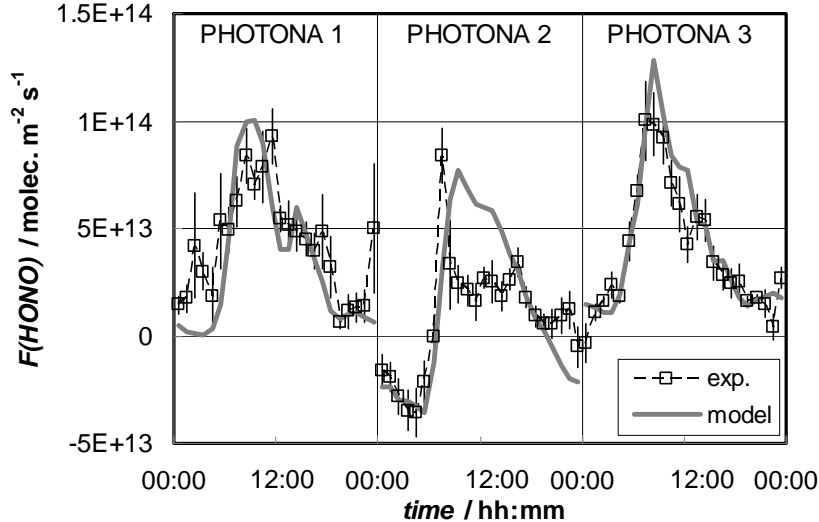

**Figure 7: Diurnally averaged measured HONO fluxes in comparison with modelled values (Eq. 8), during**

**the three PHOTONA campaigns.**