# Peer review of "Diurnal fluxes of HONO above a crop rotation"

_Atmospheric Chemistry and Physics, 2016_

## Referee Comment (RC1) · Anonymous Referee #1 · 6 Jan 2017

This is an interesting manuscript; it presents some valuable information and discussion on HONO sources over an agricultural field site. It is suitable for publication in ACP. Here are my general comments and questions:

It is surprising that microbial nitrite formation in the soil is only a minor contribution to the overall HONO emission from the ground; it is expected to be a major one over actively farmed and heavily fertilized areas such as the study site. Were soil acidity/alkalinity and nitrite content measured during the campaign? Soil (and the ground surface) acidity/alkalinity is one of the most important factors controlling the direction and the rate of HONO exchange between the air and the soil (and the surface).

The authors should assess and discuss the contribution from ground emission to the overall HONO budget in the atmospheric surface boundary layer. Based on my very rough calculation, this contribution is $\sim$30% at the noontime, assuming [HONO] $\sim$ 200 pptV, J(HONO) $\sim$ 1.1$\times$10-3 s-1 ($\sim$15 min photolysis lifetime), F(HONO) $\sim$ 5$\times$1013

molec m-2 s-1, and a surface mixed layer height of ~30 m. The vertical mixing is enhanced by surface heating during the day in summer, and HONO emitted from the ground surface may be transported up to several hundreds of meters above the ground level within its photolysis lifetime.

The measurement data of each campaign were lumped into 24 1-hr diurnal averages and then the fluxes were calculated. The authors argue that this averaging process reduced the errors of measurements for each parameter. However, a lot of detailed and valuable information was lost. If the data were averaged for each 1-hr interval, not lumped over the whole campaign, the authors may not need to filter out those "high-noise" data points and may be able to see real changes in HONO exchange direction and magnitude with many environmental factors during different events (e.g., rainy vs sunny, clean periods vs pollution episodes, . . .).

Specific comments:

P13, L475: "nigh-time" should be "nighttime".

Figs. 3 and 4, HONO gradient: HONO concentrations were measured at two heights; which one is shown in the figures? One important parameter in HONO flux calculation is the difference in HONO concentrations at the two heights ($\Delta$[HONO]). Please plot the concentrations at both heights or the $\Delta$[HONO]. The precision of F(HONO) is directly dependent on how significant the difference between the two concentrations; the difference between two similar numbers would result in a small number with a large relative uncertainty (i.e., $\sigma$(gradient) » $\Delta$[HONO]). Readers need the information to assess the accuracy of the calculated F(HONO).

Fig. 6: Several high F(HONO) data points for the morning hours should probably removed, since they may be caused by the release of trapped nitrite in dew (p 14, L 500-503). The removal of these odd data points seems to significantly improve the correlation between F(HONO) and T(soil). Would the improved F(HONO) - T(soil) correlation suggest that soil emission (from microbial nitrite formation) may be more

important after all?

Fig. 6 caption: check the equation; a left bracket is missing.

Supplemental, L7-10: Why both equations (S1) and (S2) are under "unstable conditions"? One should be for stable and the other for unstable conditions. Please cite the reference for each condition.

---

## Author Comment (AC1) · 1 Feb 2017

This is an interesting manuscript; it presents some valuable information and discussion on HONO sources over an agricultural field site. It is suitable for publication in ACP. Here are my general comments and questions:

We would like to thank Referee #1 for his/her interest and comments to the manuscript which are addressed below:

Referee #1:
It is surprising that microbial nitrite formation in the soil is only a minor contribution to the overall HONO emission from the ground; it is expected to be a major one over actively farmed and heavily fertilized areas such as the study site. Were soil acidity/alkalinity and nitrite content measured during the campaign? Soil (and the ground surface) acidity/alkalinity is one of the most important factors controlling the direction and the rate of HONO exchange between the air and the soil (and the surface).

Answer:
During the individual campaigns soil nitrite and pH were unfortunately not measured, since the microbial soil source was still not discussed at the time of the PHOTONA campaigns. However, later for the study of Oswald et al. (2013) soil samples from the PHOTONA field site were collected and analyzed for nitrite, pH and optimum HONO fluxes in the lab. This lab data, parameterized for humidity and temperature by Oswald et al. (2013), was used here to calculate potential HONO fluxes for the individual PHOTONA conditions, which were of the same order of magnitude of measured HONO fluxes only during PHOTONA 1, but were completely negligible for the other two campaigns (see section 3.2). Reasons for the variable theoretical HONO emissions by the soil source are the extremely dry conditions required for the optimum HONO fluxes based on the lab experiments of Oswald et al.. Low soil water content (SWC) was partially present only during PHOTONA 1 (and even here the SWC was still a factor of two higher compared to the optimum humidity in Oswald et al.). In contrast during PHOTONA 2 and 3 the soil was too humid to allow any significant HONO emissions from bulk soil processes. In contrast to these theoretical estimations, experimental HONO fluxes were quite comparable, especially during the two summer campaigns PHOTONA 1 and 3. Since it is unreasonable that only the soil source was the main reason for the observed HONO fluxes in PHOTONA 1, while only other sources were active during the other two campaigns, and since the correlation results are in contradiction to this source, also for the dry PHOTONA 1 campaign, we do not think that microbial nitrite production was the major origin for the observed HONO fluxes (see also the detailed discussion in section 3.4).
It should by highlighted that in the lab experiments of Oswald et al. soil samples are flushed by completely dry air which is a quite different situation compared to a typical humid atmosphere (30-100 % r.h.). Thus real HONO fluxes by the microbial soil source in a humid atmosphere might be much lower compared to these lab derived optimum HONO fluxes. Here more realistic lab experiments under atmospheric conditions are required for the future. In conclusion, the experimental field results from the present study do not confirm the proposed microbial HONO source, similar to other field studies, see e.g. Oswald et al. (2015).

Referee #1:
The authors should assess and discuss the contribution from ground emission to the overall HONO budget in the atmospheric surface boundary layer. Based on my very rough calculation, this contribution is ~30% at the noontime, assuming [HONO] ~200 pptV, $J(HONO)$ ~$1.1 \times 10^{-3}$ $s^{-1}$ (~15 min photolysis lifetime), $F(HONO)$ ~$5 \times 10^{13}$ molec $m^{-2}$ $s^{-1}$, and a surface mixed layer height of ~30 m. The vertical mixing is enhanced by surface heating during the day in summer, and HONO emitted from the ground surface may be transported up to several hundreds of meters above the ground level within its photolysis lifetime.

Answer:
The suggested calculation would be only possible using a 1-D vertical chemical transport model including meteorological information (e.g. height-dependent vertical mixing in the boundary layer, BLH, etc.) and additional chemical data (e.g. OH radical vertical profiles to calculate NO+OH contribution to the HONO levels (PSS) and vertical HONO profiles in the BL), which we do not have. Without such information calculations on the contribution of the ground source to HONO levels in the BL would be highly speculative (and height-dependent, contribution will gradually decrease to zero with increasing height…) and are in addition also out of the scope of this experimental field study.

Referee #1:
The measurement data of each campaign were lumped into 24 1-hr diurnal averages and then the fluxes were calculated.
Answer:
Our average campaign flux data used for the correlation analysis was determined differently than the referee concerned. First, 30 min (PHOTONA 1 and 2) or 5 min (PHOTONA 3) averaged campaign flux data was calculated and evaluated for potential correlations. Since this was not very successful (see below), secondly, single 24 h average campaign days were derived for each campaign after filtering the campaign data for untypical events (rain, high pollution plumes, see section 2.4).
To clarify the averaging procedure, we modified in section 2.4 (Data treatment):
"To interpret the flux data for each measurement campaign, first 30 min (PHOTONA 1 and 2) or 5 min (PHOTONA 3) averages were formed from the measurement data including the HONO fluxes. Secondly, for each campaign a diurnal average day using all this averaged data was calculated by the formation of one-hour means from the whole measurement period."

Referee #1:
The authors argue that this averaging process reduced the errors of measurements for each parameter. However, a lot of detailed and valuable information was lost. If the data were averaged for each 1-hr interval, not lumped over the whole campaign, the authors may not need to filter out those "highnoise" data points and may be able to see real changes in HONO exchange direction and magnitude with many environmental factors during different events (e.g., rainy vs sunny, clean periods vs pollution episodes,…).
Answer:
Here the referee is generally right that the analysis of individual variable diurnal data could have gained deeper information on the HONO sources. However, we decided not to use the individual data caused by several reasons:
a) There were many gaps in the individual data for which all instruments were simultaneous in operation (different zeroing, calibrations, malfunctions, etc.). Thus, in potential diurnal correlations the data coverage would have been different from day to day (e.g. start and end time of the available data, no complete rainy/sunny days, etc.) which would lead to a comparison of apples and oranges. In contrast, for the used average diurnal day (using all simultaneous campaign data) full 24 h data is available for each campaign.
b) The collected vertical gradient data showed low stationarity, which is an important criterion to analyze individual gradient data. A test for stationarity as described by Foken and Wichura (Agricultural and Forest Meteorology, 78, 83–105, 1996) showed that up to 80 % of the flux data were collected under non-stationary conditions. Here the precision of individual flux data for short specific periods (rainy/sunny/polluted) would be not very high and would not allow the interpretation of these specific short events, even if complete diurnal data were available (see a). In contrast, the use of the campaign averaged diurnal day significantly reduced the scatter of the data and showed more precise trends of the diurnal HONO flux and its correlations with $J(NO_2)$ and $NO_2$.
c) Although for short periods different processes may have been active, the aim of this study was to identify major processes describing our average daytime flux data, for which a photosensitized conversion of $NO_2$ is a more reasonable candidate than e.g. the microbial soil source, although the latter may still have been active with a much smaller average contribution or for short individual periods (see discussion in section 3.4).

Specific comments:
Referee #1:
P13, L475: "nigh-time" should be "nighttime".
Answer:
Changed, but we used the form with the hyphen, similar to the rest of the text. In addition, we changed night time to night-time in line 260.

Referee #1:
Figs. 3 and 4, HONO gradient: HONO concentrations were measured at two heights; which one is shown in the figures? One important parameter in HONO flux calculation is the difference in HONO concentrations at the two heights ($\Delta$[HONO]). Please plot the concentrations at both heights or the $\Delta$[HONO]. The precision of F(HONO) is directly dependent on how significant the difference between the two concentrations; the difference between two similar numbers would result in a small number with a large relative uncertainty (i.e., $\Delta$(gradient) » $\Delta$[HONO]). Readers need the information to assess the accuracy of the calculated F(HONO).

Answer:

In Figure 3, where the complete campaign data are shown, the upper level data is presented. Here mainly the range of situations should be presented in a similar way like in other typical HONO field campaigns, where HONO and potential precursors are collected typically in a few meters altitude above the ground. In contrast, for the correlation analysis for which the campaign average days are used (see Figures 4-7) the lower level data is presented. Using this data, source processes which are proposed to take place on the ground surfaces are better described. The exception is PHOTONA 3, where $NO_x$ was measured only in one altitude (see experimental section). We will specify the measurement levels in the revised manuscript.

In addition, we will add all $\Delta$[HONO] data including their precision errors in a modified Figure 3 (whole campaign data). In contrast in Figure 4 (average days), presentation of campaign averaged $\Delta$[HONO] data (and their errors) would not be directly comparable to the shown precision errors of the campaign averaged HONO fluxes, since those fluxes were not calculated from the campaign averaged $\Delta$[HONO] data, but present the averages of the individual fluxes over the whole campaign (see explanation to the concern above). The size of the flux errors were typically much lower than the fluxes and thus, the observed trends are statistically significant (see flux error bars in the figures).

Referee #1:

Fig. 6: Several high F(HONO) data points for the morning hours should probably removed, since they may be caused by the release of trapped nitrite in dew (p 14, L 500-503). The removal of these odd data points seems to significantly improve the correlation between F(HONO) and T(soil). Would the improved F(HONO) - T(soil) correlation suggest that soil emission (from microbial nitrite formation) may be more important after all?

Answer:

Since the controlling influence of F(HONO) by $J(NO_2)$ and $NO_2$ (see Fig. 5 and Tab. 1) is not considered in Fig. 6, the scatter of a simple plot of F(HONO) against only the temperature is obviously high. Thus, the high F(HONO) data in Fig. 6, e.g. for PHOTONA 3, are no "outliers" (by any dew evaporation) but simply reflect the different $NO_2$ and radiation levels for the same temperature (compare Figure 7, where these controlling factors are included). In contrast, the correlation with $J(NO_2) \times NO_2$ was excellent for PHOTONA 3 (see Tab. 1). In addition, only the one high F(HONO) point for PHOTONA 2 (ca. $8 \times 10^{13}$ molecules $m^{-2}$ $s^{-1}$, see Fig. 6 and Fig. 4) was explained here by dew evaporation at the very low morning temperatures of this spring campaign (see cited discussion). However, the removal of only this single point will not change too much the regression results using all data. Furthermore, since dew formation was not quantitatively studied (so the explanation of the morning peak during PHOTONA 2 is speculative and only based on results from other field studies) it would be completely unclear which data points have to be removed (by different filtering we could get almost any slope in Figure 6…). Finally, since we later described all flux data (including the night) by the equation (8) – using also a temperature dependent term – we decided to use all data in Figure 6.

In contrast, for the correlation results of the individual campaign data (see Fig. 5 and Tab. 1) the morning peak during PHOTONA 2 was already removed from the daytime data as suggested by the referee (here only the data 8:00-20:00 was used while it was 6:00-20:00 in PHOTONA 1 and 3, see text). Thus, the correlation results were already filtered for this unusual high morning peak during PHOTONA 2 and still the results point to a radiation and $NO_2$ dependent source.

Referee #1:

Fig. 6 caption: check the equation; a left bracket is missing.

Answer:

Thanks for pointing to this typo. But one bracket was deleted and not added; compare the equation in the figure.

Referee #1:

Supplemental, L7-10: Why both equations (S1) and (S2) are under "unstable conditions"? One should be for stable and the other for unstable conditions. Please cite the reference for each condition.

Answer:

The typo was corrected and the two references are valid for both cases which will be clarified in the revised manuscript. In addition, we corrected equation (S2) for another typo:

$$\Psi_{(z-d)/L} = 2 \cdot ln \left[ \frac{1 + \sqrt{1 - 16 \cdot \frac{(z-d)}{L}}}{2} \right] \tag{S2.}$$

---

## Referee Comment (RC2) · Anonymous Referee #2 · 12 Feb 2017

This paper describes three field investigations into the surface emission fluxes of nitrous acid (HONO) above soil / low crops in France. This topic relates to a series of recent studies which have demonstrated that additional fluxes of HONO to the boundary layer (beyond gas phase / heterogeneous reactions and dark NO2/H2O interactions) are required to explain observed daytime steady-state HONO levels. This is of importance as HONO is an important precursor to OH in many continental boundary layer settings, and the paper addresses a current high profile topic in atmospheric chemistry using state-of-the-science approaches.

HONO fluxes were measured using the gradient method, employing a pair of LOPAP monitors sampling at different heights above the soil / crops. The resulting fluxes are found to be comparable in magnitude to those inferred previously (although HONO flux observations are few and far between), and significantly larger than those which would

be inferred by the temperature and soil moisture dependence of biotic emissions from previous laboratory studies (but see my comments below).

The correlation between the observed flux and some potential controlling factors are observed, and the best correlation obtained for NO2, photolysis (jNO2), and temperature, adjusted for RH. This result is qualitatively attributed to photoenhanced NO2 conversion to HONO upon humic-acid type surfaces.

The paper is well written (a few minor language suggestions are given below) and clearly phrased / easy to follow. The experiments are clearly described and analysed (NB suggestions for a couple of expanded explanations below) and the conclusions, although correlation rather than definitive causation, are reasonable and advance our understanding. Subject to the points below being satisfactorily addressed, I recommend publication in ACP.

Minor points

L48 PSS also fails where there are significant heterogeneity in the co-reactants of the species in the PSS – notable OH – i.e. significant heterogeneity in VOC loading causes problems for HONO PSS analyses, even if the NO and HONO components are in a homogeneous environment.

L103 the key for PNA (HNO4) interference in the flux measurements is not the absolute amount but the different in PNA across the flux measurement heights. Can the authors comment on this.

L111 It is not appropriate to "ignore" the potential interference – please calculate (estimate) the anticipated PAN levels making reasonable assumptions and hence quantify the potential interference in the NO2.

Section 2.3 – how well was the stability criterion satisfied – what fraction of the data had to be discarded ?

L166 the diurnal averaging will address precision but not accuracy – please clarify.

Please give some more details of the "events" which were excluded – what proportion of the total were they, what criteria were used to identify them as abnormal.

L190 how often were the LOPAPs intercompared – please give details. This is critical to the flux derivations / to be confident no drift in instrument response biased the results

L313 the Oswald data derived from lab experiments in which "transient" HONO fluxes were derived as soil was dried – i.e. they would have sampled an immediate response to the changing conditions, over a period of a few hours, potentially different from the field in which conditions were much more stable. Also the samples were previously dried and reconstituted (not intact cores). Does this affect the comparison / conclusion ? Given the temperature link described subsequently – also a possible indicator of biotic influences ?

L330 were any other parameters considered in the correlations – in particular aerosol loading (ideally aerosol surface area) ?

L339 still not clear – a little unsatisfactory

L403 HNO3 sometimes shows a diurnal profile with a maximum in the afternoon as inferred here, but quite different mean diurnal profiles have also been reported (e.g. Murphy et al ACP 6 5321 2006) – which would affect the nitrate photolysis conclusion here.

L459 I wasn't quite clear how the reference RH aspect worked for the data or the fitting – please expand / clarify. May be useful to show the regression (in addition to the mean diurnal timeseries for each campaign).

Wording

Abstract line 16 – suggest reword to "...these results are consistent with HONO formation by..."

Line 38 HO2xH2O is not a nomenclature I am familiar with – use a period . ?

L55-56 reword the REA phrase

L61 an NO2 driven mechanism

L152 the abbreviations for previous decades reads a little awkwardly

---

## Referee Comment (RC3) · Anonymous Referee #3 · 22 Feb 2017

**Comment on " Diurnal fluxes of HONO above a crop rotation " by Laufs et al**

This paper presented results of flux measurements of HONO over an agriculture field using the gradient method. Based on the averaged diurnal profile, the authors calculated the HONO flux which was then used in correlation studies to explore the contributing mechanism. Photosensitized heterogeneous conversion of NO2 on soil surfaces were suggested as the major contributor to the HONO flux based on the correlation results and a local parameterization of HONO flux was also proposed. Overall, this is a well-designed study trying to answer the challenging question about atmospheric HONO source. However, there are some important issues that need to be addressed before it is accepted for publication in ACP.

**Major issues:**

**(1) HONO flux calculation**

According to Eq (2), the calculated flux depends on the gradient of HONO at two heights. If the difference between such values is too small, e.g. comparable to the systematic difference of instruments ~2-13% as shown in Fig. 2, it may invalidate most discussions. I would like to see the difference in both absolute and relative term (ΔHONO and ΔHONO/HONO)

$$F_{z_{ref}} = -\kappa \cdot u_* \cdot \frac{\partial c(HONO)}{\partial [\ln(z-d) - \Psi_{(z-d)/L}]}$$

The authors used averaged values to interpret the flux data: "*To interpret the flux data for each measurement campaign, a diurnal average was calculated by the formation of one-hour means from the whole measurement period.*" My question is how did you average $\Psi_{(z-d)/L}$ and $\Psi_{(z-d)/L}$ , etc, since their expression is different between stable and unstable conditions. In principle, it is a question whether we should first calculate the flux at each time and then do the averaging, or first do the averaging of individual parameters and then calculate the flux. Can the authors address this issue and try to make calculation for both cases?

My last question is if the effect of chemistry can be neglected in the calculation of HONO fluxes. We can make a simple HONO budget expression around noon time as follows, $\partial HONO/\partial t = \partial F/\partial z + S$
in which the change of HONO concentration ($\partial HONO/\partial t$) is subject to the gradient of flux ($\partial F/\partial z$) and the photolytic loss term (S). If $\partial HONO/\partial t \ll S$, then the contribution of S should be comparable to flux ($\partial F/\partial z$) and cannot be neglected.

**(2) Soil surface emission**

The diurnal course of soil temperature strongly depends on the depth. Here the authors used soil temperature at 5 cm in their calculations. I would suggest using soil surface temperature as in Su et al. (2011) which is more relevant for soil-atmosphere exchange. Figure 1 of Su et al. also suggests that HONO produced by photo-sensitize reaction (on the surface) is subject

to the temperature dependent equilibrium. Since the peak of soil surface temperature appears earlier than that of deeper soil (see the following figure, Jury and Horton 2004), the correlation with HONO fluxes might be improved.

[Figure]

**Figure 5.17** Diurnal variations in temperature measured at different depths in a loam soil. (After Yakuwa, 1945.)

**Minor comments:**

Ln 22: "unusually high" suggests that the measured values is higher than the expected values. Many of the references, however, don't really have an expected value from modeling or budget analysis. Thus I suggest modifying the text or limiting the references to those with budget analysis. The following references should be included into the reference list (Su et al 2008, Li et al. 2012, Yang et al. 2014).

Ln 30: "bacterial production of nitrite in soil", it is better to say "biogenic production of nitrite in soil"

Ln 42: " calculated daytime HONO sources, determined from HONO levels exceeding theoretical photostationary state (PSS) values, showed high correlations with the photolysis rate coefficient *J(NO2)* or the irradiance and NO2 concentration (Elshorbany et al., 2009; Sörgel et al. 2011b; Villena et al., 2011; Wong et al., 2012; Lee et al., 2016)." So far as I know, Su et al. (2008) is the first study performing such correlation analysis and is unfortunately missing from the reference list.

**Reference:**

W. A. Jury, R. Horton, *Soil Physics*. (Wiley, ed. 6th, 2004).
Li, X., Brauers, T., Häseler, R., Bohn, B., Fuchs, H., Hofzumahaus, A., Holland, F., Lou, S., Lu, K. D., Rohrer, F., Hu, M., Zeng, L. M., Zhang, Y. H., Garland, R. M., Su, H., Nowak, A., Wiedensohler, A., Takegawa, N., Shao, M., and Wahner, A.: Exploring the atmospheric chemistry of nitrous acid (HONO) at a rural site in Southern China, Atmos. Chem. Phys., 12, 1497-1513, 10.5194/acp-12-1497-2012, 2012.
Su, H., Cheng, Y. F., Shao, M., Gao, D. F., Yu, Z. Y., Zeng, L. M., Slanina, J., Zhang, Y. H., and Wiedensohler, A.: Nitrous acid (HONO) and its daytime sources at a rural site during the 2004 PRIDE-PRD experiment in China, JGR, 113, 10.1029/2007jd009060, 2008.

Yang, Q., Su, H., Li, X., Cheng, Y., Lu, K., Cheng, P., Gu, J., Guo, S., Hu, M., Zeng, L., Zhu, T., and Zhang, Y.: Daytime HONO formation in the suburban area of the megacity Beijing, China, Science China Chemistry, 57, 1032-1042, 10.1007/s11426-013-5044-0, 2014.

---

## Author Comment (AC2) · 19 Apr 2017

This paper describes three field investigations into the surface emission fluxes of nitrous acid (HONO) above soil / low crops in France. This topic relates to a series of recent studies which have demonstrated that additional fluxes of HONO to the boundary layer (beyond gas phase / heterogeneous reactions and dark $NO_2/H_2O$ interactions) are required to explain observed daytime steady-state HONO levels. This is of importance as HONO is an important precursor to OH in many continental boundary layer settings, and the paper addresses a current high profile topic in atmospheric chemistry using state-of-the-science approaches.

HONO fluxes were measured using the gradient method, employing a pair of LOPAP monitors sampling at different heights above the soil / crops. The resulting fluxes are found to be comparable in magnitude to those inferred previously (although HONO flux observations are few and far between), and significantly larger than those which would be inferred by the temperature and soil moisture dependence of biotic emissions from previous laboratory studies (but see my comments below).

The correlation between the observed flux and some potential controlling factors are observed, and the best correlation obtained for $NO_2$, photolysis ($jNO_2$), and temperature, adjusted for RH. This result is qualitatively attributed to photoenhanced $NO_2$ conversion to HONO upon humic-acid type surfaces.

The paper is well written (a few minor language suggestions are given below) and clearly phrased / easy to follow. The experiments are clearly described and analysed (NB suggestions for a couple of expanded explanations below) and the conclusions, although correlation rather than definitive causation, are reasonable and advance our understanding. Subject to the points below being satisfactorily addressed, I recommend publication in ACP.

We would like to thank Referee #2 for his/her interest and comments to the manuscript, which are addressed below:

Referee #2
Minor points
L48 PSS also fails where there are significant heterogeneity in the co-reactants of the species in the PSS – notable OH – i.e. significant heterogeneity in VOC loading causes problems for HONO PSS analyses, even if the NO and HONO components are in a homogeneous environment.
Answer:
Typically, when calculating a missing HONO source by the PSS approach, only the photolysis of HONO, its reaction with OH and the back reaction NO+OH are considered (some studies in addition also consider the dark conversion of $NO_2$ on surfaces). To calculate the theoretical HONO steady state concentration only J(HONO), k(NO+OH), NO and OH (and $NO_2$) are necessary. From the difference of the measured HONO to this PSS concentration a missing HONO source is calculated. Since the lifetime of the OH radical is less than a second, heterogeneity in the OH is not an issue (for OH the PSS approach is perfect…) and only heterogeneity of HONO and NO (and $NO_2$) add a significant uncertainty. Thus, any heterogeneity in the VOC loading is not an issue. The VOC loading certainly affects OH, but this is perfectly accounted for when using measured OH. In conclusion, we feel that the uncertainties of the PSS approach are well summarized by the cited study of Lee et al. (2013). For completeness we have added the very recent paper by Crilley et al. (Faraday Discussion 189, 191, 2016) as another reference to this topic.

Referee #2
L103 the key for PNA ($HNO_4$) interference in the flux measurements is not the absolute amount but the different in PNA across the flux measurement heights. Can the authors comment on this.
Answer:
We totally agree. Since the HONO flux is calculated from the difference of the HONO signals of both LOPAP instruments only the difference in a potential $HNO_4$ interference between both instruments will be important. Thus, the stated 15% interference of the upper limit $HNO_4$ level of <50 ppt (<7.5 ppt interference) is an upper limit, since it is not expected that the $HNO_4$ level at the lower LOPAP will decrease by dry deposition to 0 % of the upper instrument. For further clarification we will add: "In addition, since HONO fluxes are calculated only from the difference of both instruments, potential $HNO_4$ interferences are not considered important in the present study."

Referee #2
L111 It is not appropriate to "ignore" the potential interference – please calculate (estimate) the anticipated PAN levels making reasonable assumptions and hence quantify the potential interference in the $NO_2$.
Answer:
Unfortunately, we (a) do not know the magnitude of the PAN interference of our Luminol $NO_2$ instrument (we do not have access to a PAN-GC…) and (b) we have not measured the VOC loading during the PHOTONA field campaigns. Thus, we are not able to calculate/model potential PAN concentrations/interferences during our field

campaigns. By the way, also the Ecophysics $NO_x$ instrument (used here for NO, but is typically also used for $NO_2$) has a significant PAN interference caused by the warm photolytic converter and its long residence time of ca. 10 s, see the recent ACP paper by Reed et al. (Atmos. Chem. Phys., 16, 4707–4724, 2016). Caused by the proximity to the Paris urban region and the expected average transport time of $NO_x$, we do not expect high overestimation of $NO_2$ by PAN interferences. And even if we had significant PAN interferences in the present study, the higher expected PAN levels in the early afternoon (photochemically formed) could not explain our maximum HONO fluxes in the morning. Thus, the qualitative result (correlation of the HONO flux with $NO_2 \times J(NO_2)$ would not have negated by such an interference. It is even the opposite, if potential PAN interferences were corrected (=> the daytime $NO_2$ profile would get than even more asymmetric, see Fig. 4) the correlation of the HONO flux with $NO_2 \times J(NO_2)$ would have increased compared to the other proposed HONO sources, which are expected to maximize in the early afternoon (see discussion in section 3.4).

Referee #2
Section 2.3 – how well was the stability criterion satisfied – what fraction of the data had to be discarded?
Answer:
The fraction of time with Obukhov length lower than 5 in absolute value was 24 %, 21 % and 16 % in 2009, 2010 and 2011, respectively, while very unstable conditions ($L < 0$) occurred 13 %, 12 % and 7 % of the time, respectively. We however did not filter the data for stability, as stable conditions also generally corresponded to small fluxes, and we rather wanted to consider as much possible of the data during each campaign to evaluate diurnal averages of the fluxes. We however filtered for events that could affect the quality of the concentration measured and which were untypical for that agricultural field site (rain, emission plumes), see section 3.4. Filtering for non-stationarity conditions was not necessary since we continuously sampled at two heights with two individual LOPAP instruments. Indeed any non-stationarity in concentrations was capture simultaneously at the two heights and did not affect the measured gradient, as opposed to methods based on successive sampling with a single instrument (e.g. Stella et al., 2011).

Referee #2
L166 the diurnal averaging will address precision but not accuracy – please clarify. Please give some more details of the "events" which were excluded – what proportion of the total were they, what criteria were used to identify them as abnormal.
Answer:
The errors used for the 30/5 min data (PHOTONA 1+2/3, see new Figure 3) are only precision errors, as for the calculation of the HONO flux mainly the difference between the two instruments are of importance (and not any systematic errors, e.g. any calibration error, similar valid for both instruments, calibrated by the same nitrite standard). In contrast, when the average day was formed (see data shown in Figures 4-7) the standard error was calculated (standard deviation divided by the square root of the numbers of values: $\sigma/\sqrt{n}$).
Concerning the filtering of the data, first of all only 52%, 77% and 78% of the campaign data could be used to determine HONO fluxes during PHOTONA 1, 2 and 3, respectively. For the other periods, data from all instruments were simultaneously not available (loss by calibrations, zeros, zero gradients, malfunctions etc.). From this flux data, 97%, 99%, and 57% were used to determine the average days for PHOTONA 1, 2 and 3, respectively. For PHOTONA 1 and 2 only few data were filtered caused by a rain event (PHOTONA 1: 24.08.) and by a high pollution plume (PHOTONA 2: 07.04.). In contrast, for PHOTONA 3, a significant fraction of the flux data was discarded (16.08.-21.08.) caused by low quality of the first intercalibration during this campaign, which caused untypical continuous negative HONO fluxes, which were not observed later during the campaign. For the average day we used only the data starting from the 21.08., when the next intercalibration was performed (see Fig. 2). Finally, data from the 26.08. were also not considered, caused by a strong rain event leading to negative HONO fluxes. This information will be added to the revised manuscript.

Referee #2
L190 how often were the LOPAPs intercompared – please give details. This is critical to the flux derivations / to be confident no drift in instrument response biased the results.
Answer:
During PHOTONA 1, 2 and 3 the two LOPAPs were intercompared 7, 3, and 3 times, respectively. Here, high stability of the instrument's responses was observed during PHOTONA 1 and 2, while higher variability between both instruments was observed at the beginning of PHOTONA 3. This latter data was however not used when generating the average day (see last concern and high fraction of discarded data during this campaign). Since the small variability of the instrument's responses is considered in the precision errors of the gradient (see equation 3), the results of the present study are not significantly biased by any instruments drifts.

Referee #2

L313 the Oswald data derived from lab experiments in which "transient" HONO fluxes were derived as soil was dried – i.e. they would have sampled an immediate response to the changing conditions, over a period of a few hours, potentially different from the field in which conditions were much more stable. Also the samples were previously dried and reconstituted (not intact cores). Does this affect the comparison / conclusion? Given the temperature link described subsequently – also a possible indicator of biotic influences?

Answer:

We agree, the experiments explained by Oswald et al. do not represent a real atmospheric situation. Thus, we already wrote in section 3.4:

"It should be stressed that in the Oswald et al. (2013) study the experimental conditions were not representative for the present field site. While in these laboratory studies the upper soil surface was flushed by completely dry air, leading to optimum HONO emissions only at very dry conditions, the relative humidity never decreased below 26 %, 31 % and 43% in PHOTONA 1, 2 and 3, respectively. More work is desirable to reconcile HONO field data with incubation experiments as performed by Oswald et al. (2013)."

However, since we do not have any other parameterization of the soil HONO source than that published by Oswald et al. we cannot comment whether any other (unknown) humidity dependence of the biogenic HONO source in a real atmosphere would affect our results. We can only conclude here that a source as explained by Oswald et al. cannot describe our field observations.

In addition also the temperature dependence of the HONO source (see Figure 6) was much weaker compared to the radiation and $NO_2$ dependence for each campaign (see table 1) and can be explained by any (…) heterogeneous HONO source and the expected temperature dependence of the HONO adsorption on soil surfaces. Also the temperature dependence was a much weaker influencing parameter compared to the soil humidity in Oswald et al. Here the similar HONO fluxes in both summer campaigns (PHOTONA 1 and 3) under the very different soil water contents clearly indicate that any bulk soil HONO sources (not necessarily biogenic…) cannot explain our field observations, see section 3.4.

Referee #2

L330 were any other parameters considered in the correlations – in particular aerosol loading (ideally aerosol surface area)?

Answer:

We only considered those parameters which were directly measured; see experimental section. Particle levels were not measured during the PHOTONA campaigns, since HONO formation on particles surfaces was not considered of importance (see e.g. Kleffmann et al., 2003). In addition, since all known particle sources (nitrate photolysis, $NO_2$+HA+hv, $NO_2$+SOA…) show the same HONO formation kinetics compared to similar ground surfaces in laboratory studies and since the S/V ratio of particles are orders of magnitude lower compared to the ground surfaces in near ground measurements, no significant HONO formation on particles is expected during the PHOTONA campaigns. The situation may be different at higher altitude, e.g. in the free troposphere, see the recent paper by Ye et al. (Nature, 532, 489-491, 2016).

Referee #2

L339 still not clear – a little unsatisfactory

Answer:

We agree; we could not identify a final main reason for the different results between the spring (PHOTONA 2) and the two summer campaigns (PHOTONA 1+3). That was the reason why we gave three potential explanations in lines 337-347.

Referee #2

L403 $HNO_3$ sometimes shows a diurnal profile with a maximum in the afternoon as inferred here, but quite different mean diurnal profiles have also been reported (e.g. Murphy et al ACP 6 5321 2006) – which would affect the nitrate photolysis conclusion here.

Answer:

We do not understand this issue, since the diurnal $HNO_3$ profiles shown in Figure 15c of Murphy et al. (red and black symbols) are perfectly in line with our statement showing the typical asymmetric $HNO_3$ profile maximizing in the afternoon. As explained in our manuscript, these profiles are a result of the main formation pathway of $HNO_3$ by $NO_2$+OH in a daytime atmosphere (especially in summer) and could not explain our HONO fluxes maximizing in the morning. If the referee considers the green data from Figure 15c of the cited paper ($HNO_3$ is constant over the day) heterogeneous photolysis of $HNO_3$ could also not explain our diurnal flux observations. Finally, we are not aware of any study in which $HNO_3$ levels maximized in the morning similar to our HONO fluxes.

Referee #2
L459 I wasn't quite clear how the reference RH aspect worked for the data or the fitting – please expand / clarify. May be useful to show the regression (in addition to the mean diurnal timeseries for each campaign).
Answer:
The used linear humidity dependence in equation (8) is not a results of any regression analysis, but was only added to take into consideration the positive influence of relative humidity on the proposed heterogeneous sources and sinks derived from laboratory studies at medium humidity ($NO_2$+surfaces/dark; $NO_2$+HA+hν, HONO deposition), see references in lines 458-459. The use of the 50 % RH reference humidity was accidentally chosen and any other humidity reference point could have also been used, resulting in exactly the same experiment/model agreement (after linearly adjusting the parameters A, B and ν(HONO)$_T$). We used here 50 % RH simply because this reflects a typical average humidity in the atmosphere. In addition, a linear humidity dependency was applied here for simplicity, although laboratory derived humidity dependencies often level off at high humidity.

Referee #2
Wording
Abstract line 16 – suggest reword to "…these results are consistent with HONO formation by…"
Answer:
Changed

Referee #2
Line 38 $HO_2xH_2O$ is not a nomenclature I am familiar with – use a period . ?
Answer:
Changed according also to the used definition of this complex in the original paper by Li et al.

Referee #2
L55-56 reword the REA phrase
Answer:
We do not understand this issue? REA is the only used abbreviation for the "relaxed eddy accumulation" method?

Referee #2
L61 an $NO_2$ driven mechanism
Answer:
Changed

Referee #2
L152 the abbreviations for previous decades reads a little awkwardly
Answer:
We change to: "During the 1960's and 1970's…"

---

## Author Comment (AC3) · 19 Apr 2017

This paper presented results of flux measurements of HONO over an agriculture field using the gradient method. Based on the averaged diurnal profile, the authors calculated the HONO flux which was then used in correlation studies to explore the contributing mechanism. Photosensitized heterogeneous conversion of $NO_2$ on soil surfaces were suggested as the major contributor to the HONO flux based on the correlation results and a local parameterization of HONO flux was also proposed. Overall, this is a well-designed study trying to answer the challenging question about atmospheric HONO source. However, there are some important issues that need to be addressed before it is accepted for publication in ACP.

We would like to thank Referee #3 for his/her interest and comments to the manuscript, which are addressed below:

Referee #3:
Major issues:
(1) HONO flux calculation
According to Eq (2), the calculated flux depends on the gradient of HONO at two heights. If the difference between such values is too small, e.g. comparable to the systematic difference of instruments ~2-13% as shown in Fig. 2, it may invalidate most discussions. I would like to see the difference in both absolute and relative term (ΔHONO and ΔHONO/HONO).

Answer:
First, the relative errors of the Δ[HONO] data induced by the agreement between both instruments will be lower than e.g. the 13 % shown for the intercalibration during PHOTONA 3 (see Figure 2), since the results from the intercalibrations were used to harmonize the LOPAP instruments (see lines 196-198). Thus, although both instruments may have a higher absolute uncertainty, after this harmonization the sign of the fluxes will be only affected by the lower precision errors of the two instruments. In addition, in the revised manuscript, we will add all Δ[HONO] data including their precision errors in a modified Figure 3 (whole campaign data). In contrast in Figure 4 (average days, from which all correlations are derived), presentation of campaign averaged Δ[HONO] data (and their errors) would not be directly comparable to the shown standard errors ($\sigma/\sqrt{n}$) of the campaign averaged HONO fluxes, since the average fluxes were not calculated from the campaign averaged Δ[HONO] data, but present the averages of the individual fluxes over the whole campaign (see also next point). The flux standard errors were typically much lower than the fluxes and thus, the observed trends are statistically significant (see flux error bars in the figures).

Referee #3:
The authors used averaged values to interpret the flux data: "*To interpret the flux data for each measurement campaign, a diurnal average was calculated by the formation of one-hour means from the whole measurement period.*" My question is how did you average and, etc, since their expression is different between stable and unstable conditions. In principle, it is a question whether we should first calculate the flux at each time and then do the averaging, or first do the averaging of individual parameters and then calculate the flux. Can the authors address this issue and try to make calculation for both cases?

Answer:
Unfortunately, this point was not described clearly enough in our manuscript (see also same question by referee #1). Here first, 30 min (PHOTONA 1 and 2) or 5 min (PHOTONA 3) averaged campaign flux data were calculated using different stability integrated functions $\Psi$ for stable and unstable conditions (see supplement). Secondly, single 24 h average campaign days (hourly data) were derived for each campaign after filtering the campaign data for untypical events (rain, high pollution plumes, see section 2.4).
To clarify the averaging procedure, we modified in section 2.4 (Data treatment):
"To interpret the flux data for each measurement campaign, first 30 min (PHOTONA 1 and 2) or 5 min (PHOTONA 3) averages were formed from the measurement data including the HONO fluxes. Secondly, for each campaign a diurnal average day using all this averaged data was calculated by the formation of one-hour means from the whole measurement period."

Referee #3:
My last question is if the effect of chemistry can be neglected in the calculation of HONO fluxes. We can make a simple HONO budget expression around noon time as follows,
$\partial HONO/\partial t = \partial F/\partial z + S$
in which the change of HONO concentration ($\partial HONO/\partial t$) is subject to the gradient of flux ($\partial F/\partial z$) and the photolytic loss term (S). If $\partial HONO/\partial t \ll S$, then the contribution of S should be comparable to flux ($\partial F/\partial z$) and cannot be neglected.

Answer:
This point is already considered in section 2.5.3 in which we exactly investigated the proposed concern. By comparing the photolytic lifetime of HONO as the fastest HONO chemical term with the transport time (see equations 4 and 5) the influence of the chemistry on the HONO fluxes were found to be less than 10 % and thus any correction for chemistry was ignored (see lines 239-243).

Referee #3:
(2) Soil surface emission
The diurnal course of soil temperature strongly depends on the depth. Here the authors used soil temperature at 5 cm in their calculations. I would suggest using soil surface temperature as in Su et al. (2011) which is more relevant for soil-atmosphere exchange. Figure 1 of Su et al. also suggests that HONO produced by photo-sensitize reaction (on the surface) is subject to the temperature dependent equilibrium. Since the peak of soil surface temperature appears earlier than that of deeper soil (see the following figure, Jury and Horton 2004), the correlation with HONO fluxes might be improved.
Answer:
We generally agree to that issue and especially temperatures at deeper soil layers show significant different diurnal profiles compared to the surface temperature. However, in our case the measured soil temperature a 5 cm depth was not too different compared to the theoretical surface temperature $T(z_{0'})$, see Figure below.

[Figure]

Especially for PHOTANA 1, the daytime profiles were almost the same and for PHOTONA 3 at least the shape and the timing of the maximum temperatures were also quite similar. Only for PHOTONA 2 the diurnal profile of $T(z_{0'})$ was shifted slightly to earlier daytime as shown by the referee by the data from the study of Jury and Horton (2004). Despite the similar shapes of the diurnal temperature profiles, especially for the two summer campaigns, we have repeated the correlation analysis using also the proposed surface temperature. As expected, the use of $T(z_{0'})$ did not improve the quality of the correlations compared to the use of the directly measured soil temperature, see blue numbers in the following table.

**Table 1: Goodness of the weighted orthogonal regressions of hourly average daytime data (6:00 to 20:00 UTC) of *F(HONO)* against different variables for the three PHOTONA campaigns. The numbers represent $\chi^2/Q$ ($R^2$) values for which lower $\chi^2$ and higher $Q$ and $R^2$ values indicate better correlations (for definition see Brauers and Finlayson-Pitts, 1997). Bold numbers represent the strongest correlations observed for each campaign. Data for the correlation using the theoretical surface temperature $T(z_{0'})$ is added in blue.**

| | $J(NO_2)$ | $J(O^1D)$ | $T_{soil}$, $T(z_{0'})$ | $u_*$ | $J(NO_2)\cdot c(NO_2)$ |
|---|---|---|---|---|---|
| **PHOTONA 1** | 27.5/0.004 (0.47) | not measured | 50.2/6·10⁻⁷ (0.22) 46.4/3·10⁻⁶ (0.21) | 23.4/0.016 (0.41) | **7.27/0.78 (0.79)** |
| **PHOTONA 2** | 12.1/0.28 (0.27) | not measured | 9.33/0.50 (0.019) 8.5/0.58 (0.29) | **5.66/0.84 (0.37)** | 12.4/0.26 (0.37) |
| **PHOTONA 3** | 53.7/3·10⁻⁷ (0.38) | 79.8/5·10⁻¹² (0.17) | 121/5·10⁻²⁰ (0.03) 91.9/2·10⁻¹⁴ (4·10⁻⁷) | 62.7/7·10⁻⁹ (0.20) | **3.26/0.994 (0.85)** |

Since the correlations of the HONO flux with $J(NO_2)\cdot c(NO_2)$ were still much better compared to those when using $T(z_{0'})$, see bold data, our conclusion is not affected by the use of the different temperatures. Since in addition $T(z_{0'})$ is a theoretical temperature showing some degree of uncertainty, we prefer using the directly measured soil temperature at the lowest depth of 5 cm.

Referee #3:
Minor comments:
Ln 22: "unusually high" suggests that the measured values is higher than the expected values. Many of the references, however, don't really have an expected value from modeling or budget analysis. Thus I suggest modifying the text or limiting the references to those with budget analysis. The following references should be included into the reference list (Su et al 2008, Li et al. 2012, Yang et al. 2014).
Answer:
We will add two of the suggested references to the revised manuscript (there are too many on this topic…), while the one by Yang et al. (2014) was already used. However, we also would like to show the long history of the proposed missing daytime HONO source starting with the first study by Neftel et al. (1996). Already in this study a clear missing daytime HONO source was identified, although the authors could not make a complete budget analysis, caused mainly by the missing measured OH (which is by the way also not available in the proposed study by Su et al.). However, whatever reasonable OH data is used in most former calculations, the measured daytime HONO would be not in balance with its known sources and sinks. E.g. in the first study in which all necessary parameters were measured to answer this issue (Kleffmann et al., 2005), a completely unreasonable, more than an order of magnitude higher OH concentration would have been necessary to get the PSS similar to the measured HONO (=> "unreasonably high"). Thus, also former studies already proved the existence of a daytime source of HONO, although with higher uncertainty in the absolute magnitude.

Referee #3:
Ln 30: "bacterial production of nitrite in soil", it is better to say "biogenic production of nitrite in soil"
Answer:
Will be changed in the revised manuscript.

Referee #3:

Ln 42: "calculated daytime HONO sources, determined from HONO levels exceeding theoretical photostationary state (PSS) values, showed high correlations with the photolysis rate coefficient $J(NO_2)$ or the irradiance and $NO_2$ concentration (Elshorbany et al., 2009; Sörgel et al. 2011b; Villena et al., 2011; Wong et al., 2012; Lee et al., 2016)." So far as I know, Su et al. (2008) is the first study performing such correlation analysis and is unfortunately missing from the reference list.

Answer:

We will add the suggested reference, which we originally not used, since $J(NO_2)$ was not measured in that study and was calculated by the TUV model, which might be uncertain under the high aerosol load and the corresponding influence of light scattering on the UVA levels under these highly hazy Chinese (Beijing) conditions. In addition, a $J(NO_2)$ dependent daytime source was already used in a former study (Vogel et al., 2003), which we now also added to the references besides the most recent one by Crilley et al. (2016).

---

## Author Response (AR2)

**BERGISCHE UNIVERSITÄT WUPPERTAL**

PD Dr. Jörg Kleffmann
Privatdozent

Department of Physical and Theoretical Chemsitry,
Faculty for Mathematics and Natural Sciences
University of Wuppertal
Gaußstr. 20, 42119 Wuppertal, Germany

| Room | H.13.06 |
|---|---|
| Phone | +49 (0)202 439 3534 |
| Fax | +49 (0)202 439 2757 |
| Mail | kleffman@uni-wuppertal.de |
| Internet | http://www.ptc.uni-wuppertal.de |
| Reference | |

| Date | 15.05.2017 |

Bergische Universität Wuppertal, PD Dr. Jörg Kleffmann,
Gaußstr. 20, 42119 Wuppertal, Germany

To
Prof. James Roberts
Editor ACP
Manuscript No.: acp-2016-1030

Dear Prof. Roberts,

please find attached our revised final manuscript "Diurnal fluxes of HONO above a crop rotation" by Laufs et al. Two of the corrections (2+3) have been applied as suggested. However, concerning your first suggestion (nighttime vs. night-time) we have used the British form (night-time) throughout the manuscript (in British English…) after discussion with the editorial office.

Best regards

Jörg Kleffmann